# ERA: Evidence-Based Reasoning and Augmentation for Open-Vocabulary Medical Vision

## Abstract

Vision-Language Models (VLMs) have shown great potential in the domain of open-vocabulary medical imaging tasks. However, their reliance on implicit correlations instead of explicit evidence leads to unreliable localization and unexplainable reasoning processes. To address these challenges, we introduce **ERA** (**E**vidence-Based **R**easoning and **A**ugmentation), a novel framework that transforms VLMs from implicit guessers into explicit reasoners for medical imaging. ERA leverages Retrieval-Augmented Generation (RAG) and Chain-of-Thought (CoT) to construct a traceable reasoning path from evidence to results. This framework requires no additional training and can be readily applied on top of any existing Vision-Language Model. Evaluated across multiple challenging medical imaging benchmarks, ERA's performance is comparable to fully-supervised specialist models and significantly surpasses current open-vocabulary baseline methods. ERA offers a promising direction for developing more auditable and transparent Vision-Language Models for medical applications.

## 1 Introduction

Prompt-based models like the Segment Anything Model (SAM) are a major step forward in image segmentation Kirillov et al. (2023). They offer great flexibility and precision by outlining objects based on user inputs. In specialized fields like medicine, however, this approach has a key limitation: it relies on manual interaction. To use these models well in a clinic, an operator needs deep medical knowledge to ensure accuracy. Also, the growing volume of diagnostic data makes a manual, case-by-case method slow and impractical. This scaling problem shows the need for methods that can automatically create spatial prompts, which is vital for using large models widely in medicine.

To automate this process, a simple idea is to train an object detector for specific medical tasks to generate prompts like bounding boxes. Yet, this method faces big challenges in getting medical data. Strict patient privacy rules, the high cost of expert annotation, and slow labeling create a severe lack of large, high-quality datasets. This data shortage makes it nearly impossible to train a robust detector for diverse, open-vocabulary needs. This problem calls for a new approach that moves away from models needing extensive in-domain training. Vision-Language Models (VLMs) are a promising alternative Feng et al. (2025); Zhang et al. (2025); Xie et al. (2025); Shen et al. (2025). Pre-trained on vast general image-text data, VLMs can understand open-vocabulary commands and perform initial localization without specialized data, helping to overcome the data shortage.

Although VLMs offer a good solution for data scarcity, two major flaws block their direct use in clinical practice and make them unreliable Zhang et al. (2025); Li et al. (2025b); Vaswani et al. (2017). First, they rely on hidden patterns. Their localization decisions often depend on unclear statistical correlations from general-domain data, not the clear medical evidence needed for accurate localization. This leads to unreliable prompts. Second, their reasoning process is a "black box" that cannot be traced. This conflicts with the clinical need for every decision to be based on verifiable evidence, making these models difficult to trust in safety-critical applications.

To address these core challenges, we propose ERA (Evidence-based Reasoning and Augmentation), a framework that transforms a VLM from an implicit guesser into an explicit reasoner. Instead of fine-tuning, ERA restructures the model's inference process. It uses RAG to find verifiable evidence from an external medical knowledge base Fan et al. (2024); Du et al. (2024); Qi et al. (2024). Subsequently, it employs a CoT to build a structured, traceable reasoning path Wang et al. (2025); Lai

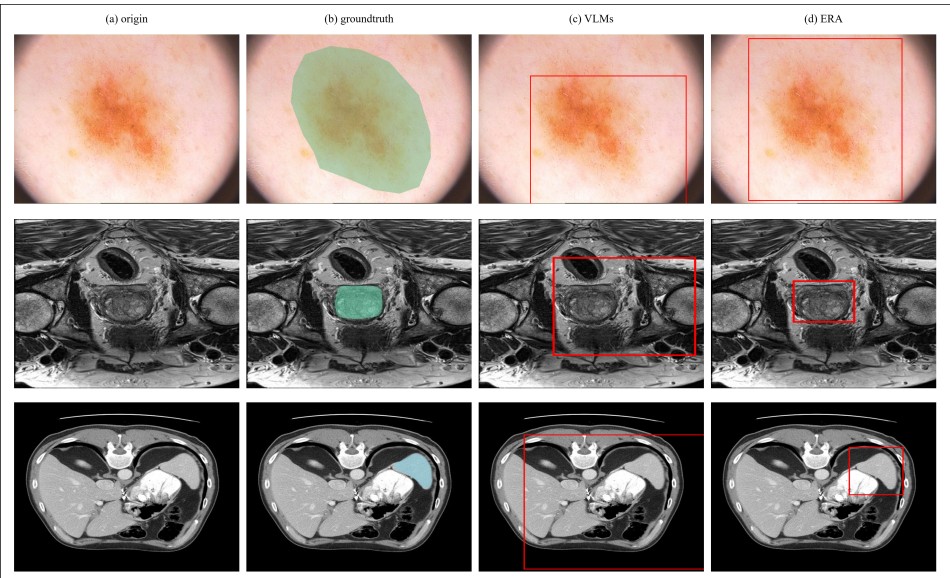

Figure 1: Visual comparison of a standard VLM versus our ERA framework on the localization task. Columns show (a) the original medical image, (b) the ground truth segmentation, (c) the localization result from a typical generalist VLM, and (d) the result from our ERA framework. Relying on opaque, implicit knowledge, the generalist VLM's localization (c) is often imprecise or overly coarse. In contrast, our ERA framework (d), by grounding its reasoning in explicit evidence, generates a significantly more precise and reliable spatial prompt that aligns closely with the ground truth.

& Nissim (2024). This mechanism guides the VLM to cross-reference retrieved evidence against the image before generating a high-confidence prompt. By documenting this reasoning chain, ERA renders the decision process auditable—an essential step toward building trust. Our work significantly improves VLM performance in the medical domain, outperforming existing open-vocabulary methods and achieving precision close to fully-supervised specialist models.

The main contributions of this paper are:

- We propose ERA, a framework that guides VLMs from unreliable guessing toward explicit, evidence-based reasoning. This approach offers a key path to improving the robustness and traceability of VLMs in medical tasks.

- We design a reasoning architecture that joins RAG with a CoT process. This synergy forces the model to ground its decisions in external, verifiable medical knowledge.

- Our framework transforms the VLM's black-box decision process into a transparent and auditable workflow. By generating a clear reasoning path, it provides a verifiable basis for decision-making in high-stakes medical settings.

- Extensive experiments show that ERA performs robustly on specialized medical datasets where other zero-shot generalist models fail completely, proving the effectiveness of our evidence-based approach.

## 2 RELATED WORK

### 2.1 SEGMENT ANYTHING MODEL 2

Prompt-based interaction has recently become a powerful paradigm in computer vision, with the SAM marking a significant milestone by demonstrating unprecedented zero-shot segmentation capabilities on a massive dataset Kirillov et al. (2023). Its successor, SAM2, further extends this zero-shot capacity from static images to the video domain, establishing a unified, promptable foundation

model for visual segmentation Ravi et al. (2024). Beyond introducing mechanisms like streaming memory for temporal data, SAM2 also surpasses the original in image segmentation, achieving higher precision and a manifold increase in speed Ravi et al. (2024); Xiong et al. (2024); Guo et al. (2025); Bai et al. (2025). Despite their formidable power, the performance of these models is fundamentally contingent on the quality of the input prompts they receive. Consequently, the challenge of reliably and automatically generating precise prompts to overcome the bottleneck of manual interaction constitutes the central problem our research aims to address.

## 2.2 VISION-LANGUAGE MODELS

To address the aforementioned prompting bottleneck, Vision-Language Models (VLMs) offer a highly promising technical pathway for automation Jang et al. (2025); Yamaguchi et al. (2025). The new generation of VLMs has moved beyond the simple image-text alignment of earlier models like CLIP, exhibiting deeper levels of vision-language fusion and reasoning. Among these, models like Qwen-2.5 Team (2024) stand out, built upon an advanced large language model deeply integrated with a powerful visual encoder. This architecture enables complex tasks ranging from detailed image description to precise referential comprehension, making them ideal candidates for generating spatial prompts from natural language Li et al. (2025a); Feng et al. (2025). However, a fundamental challenge persists even with these powerful VLMs: their decision-making process relies on implicit statistical correlations learned from general-domain data, not on the explicit, evidence-based reasoning essential for medical diagnostics Zhang et al. (2025); Li et al. (2025b). This inherent limitation is precisely the target our ERA framework is designed to resolve.

## 2.3 RETRIEVAL-AUGMENTED GENERATION

To address the VLM's lack of explicit evidence, our framework turns to RAG, a pivotal paradigm from Natural Language Processing (NLP) Fan et al. (2024). The core principle of RAG is to retrieve relevant information from a large-scale, trusted external knowledge base to serve as context before a model proceeds with generation or reasoning. By grounding decisions in external, verifiable knowledge, this mechanism has been proven to effectively reduce model hallucinations and enhance the factual accuracy of generated content Zhang et al. (2025). In this work, we adapt the RAG paradigm to the task of visual localization, providing the VLM with the explicit evidential foundation it inherently lacks. This approach equips the model with a reliable external reference, systematically solving its predicament of relying on vague internal knowledge and implicit guesswork for its conclusions.

## 2.4 CHAIN OF THOUGHT

While RAG provides the necessary evidence, CoT provides the mechanism to ensure this evidence is used in a traceable and rigorous manner Wang et al. (2025). Inspired by the Chain of Thought concept, CoT guides a model to generate a series of intermediate, step-by-step logical inferences before arriving at a final answer Liang et al. (2025). This structured approach not only boosts performance on complex tasks but also significantly enhances model interpretability by exposing the reasoning process. Within our framework, CoT serves not for general-purpose reasoning but for the specific purpose of constructing an explicit and traceable validation path. This path makes the VLM's process of adopting external evidence both rigorous and auditable, providing a logical guarantee for high-reliability prompts and directly addressing the fundamental demand in clinical applications for trustworthy, evidence-based decision-making.

# 3 METHOD

## 3.1 OVERALL FRAMEWORK

To solve the problem of VLMs relying on unclear, internal knowledge for important medical tasks, we introduce ERA (Evidence-based Reasoning and Augmentation). ERA is a framework made to enforce a clear, evidence-based reasoning process. As shown in Figure 2, ERA changes a pretrained VLM from a simple guesser into a careful reasoner. It does this by connecting the VLM to

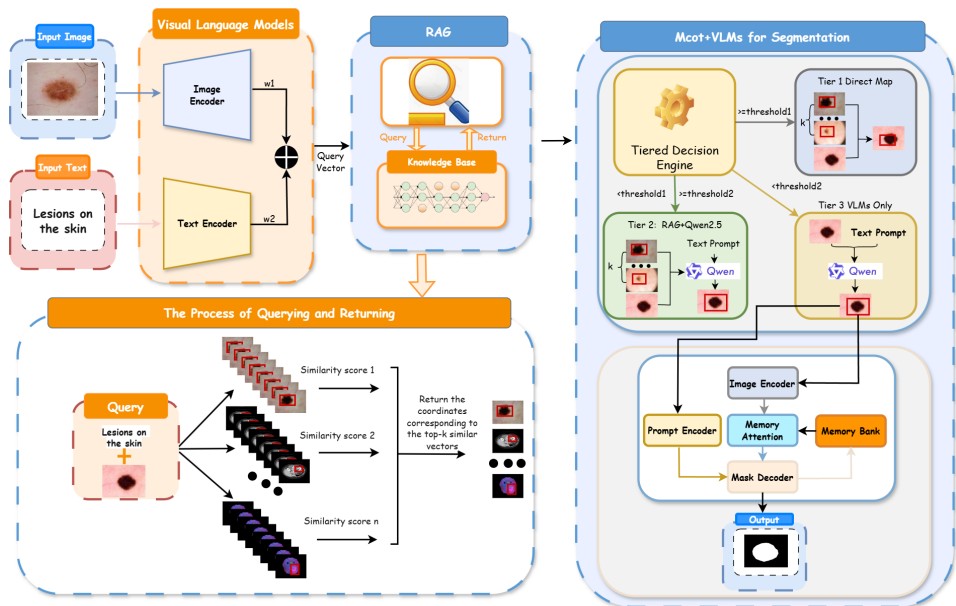

Figure 2: Overview of the ERA Framework. Given an input image and a text instruction, a query vector is formed by a visual language model. This vector is used to retrieve the most relevant visual exemplar from a pre-computed knowledge base to serve as evidence. Subsequently, the input image, text, and the retrieved exemplar are fed into the core Deliberative Reasoning Engine. The engine executes a tiered decision policy guided by a Chain-of-Thought to validate the evidence and synthesize a final, high-confidence spatial prompt, which is then used to drive a segmentation model.

an external, non-parametric medical knowledge base. The framework operates in a zero-shot manner, requiring no task-specific training. ERA remains effective with any value arbitrarily selected from the reference ranges 7. Instead, it guides the VLM's existing abilities through a structured and checkable reasoning process. This process has two main parts: a Non-parametric Knowledge Integration module to find real-world evidence, and a Deliberative Reasoning Engine to check and use that evidence.

## 3.2 THE NON-PARAMETRIC KNOWLEDGE BASE

To allow for clear reasoning, our framework uses an external, non-parametric visual knowledge base. We build and index this base to provide checkable evidence that adds to the VLM's own understanding.

### 3.2.1 KNOWLEDGE CURATION AND STRUCTURING

At the center of our framework is a large, structured medical knowledge base, $\mathcal{K}$, which serves as the source of verifiable evidence. We construct this base using MedIMeta Woerner et al. (2025), a comprehensive standardized meta-dataset that aggregates high-quality medical images and ground-truth annotations from 10 diverse medical domains, including CT, MRI, X-ray, and dermatoscopy. We employ this composite, mixed-modality source specifically to maximize the diversity of visual evidence. Unlike single-domain databases that restrict a model to specific anatomies, a composite database ensures that the retrieval module can find semantically relevant visual analogues even for rare or unseen clinical targets, thereby preventing overfitting to a narrow subdomain.

Each item $e \in \mathcal{K}$ is structured as a tuple $e = (i, t, b)$. Here, $i$ is the image path, and $t$ is the textual label. Crucially, since the source datasets primarily provided pixel-level segmentation masks, we algorithmically generated the spatial coordinate $b$ by calculating the **minimum bounding rectangle** for each mask. This ensures that the geometric anchor tightly frames the target lesion or organ, converting diverse segmentation data into a unified prompt format compatible with our framework.

### 3.2.2 FEATURE SPACE INDEXING FOR EFFICIENT RETRIEVAL

To allow for fast, meaning-based evidence retrieval, the entire knowledge base $\mathcal{K}$ is indexed beforehand. This one-time pre-calculation of features makes the retrieval process as fast as possible during use. We use a pre-trained vision-language model, BLIP2 Li et al. (2023), as a feature encoder that is not changed. We chose BLIP2 because it is good at understanding meaning and works well on new data. Using this encoder, each image in $\mathcal{K}$ is turned into a feature vector in a high-dimensional space, which is then normalized. This normalization ensures that the inner product of any two vectors equals their cosine similarity. It is worth noting that despite the varying resolutions and aspect ratios of the source images in the heterogeneous knowledge base, the image encoder standardizes all inputs via resizing and padding during the embedding process, ensuring consistent feature representation across different modalities.

During use, a query made of an image $I$ and a text instruction $C$ is encoded into a normalized query vector. The top-$k$ most similar items from the knowledge base are then found by an efficient inner product calculation. This retrieval process is written as:

$$E_{\text{cand}} = \text{Retrieve}(I, C; \mathcal{K}) \tag{1}$$

where $E_{\text{cand}}$ is the set of candidate examples found in the knowledge base $\mathcal{K}$ based on the query $(I, C)$.

## 3.3 THE DELIBERATIVE REASONING ENGINE

Finding relevant evidence is only the first step. The key innovation of ERA is its careful process for using that evidence. This module uses a powerful, standard VLM as its reasoning core, which we call $\Phi$. It guides the VLM's behavior with a carefully designed CoT to check and use the retrieved evidence in a structured, traceable way.

### 3.3.1 THE PARAMETRIC REASONING CORE

The core of our reasoning engine is Qwen2.5, a powerful, open-source Vision-Language Model. We use its advanced abilities in a zero-shot setting, treating it as a general-purpose reasoner $\Phi$. To make it run efficiently, we use methods like 8-bit quantization and Flash Attention 2. This allows the framework to work well without needing costly fine-tuning.

### 3.3.2 CHAIN-OF-THOUGHT FOR EVIDENCE-BASED REASONING

To guide the VLM's reasoning, we designed a confidence-aware tiered CoT prompting strategy. The framework retrieves a candidate set $E_{\text{cand}}$ containing $k$ exemplars ($k = 6$). The routing logic is determined by the similarity score of the top-ranked exemplar ($E^* \in E_{\text{cand}}$). If the similarity exceeds a high-confidence threshold, the system enters Tier 1, efficiently adopting $E^*$ without further computation. However, if the confidence is ambiguous, the system enters Tier 2. In this mode, unlike Tier 1, the VLM reasoner $\Phi$ is fed the entire candidate set $E_{\text{cand}}$ as reference context. This allows the model to cross-reference multiple pieces of evidence to robustly validate the primary candidate before executing the following logical steps:

1. **Step 1: Check for Concept Match.** The reasoner $\Phi$ analyzes the context provided by $E_{\text{cand}}$ to determine if the primary evidence $E^*$ is semantically relevant. It judges whether the visual features in the evidence align with the target description in instruction $C$, yielding a judgment $v_c \in \{\text{True}, \text{False}\}$.

2. **Step 2: Test the Location Hypothesis.** Only if the concepts match ($v_c = \text{True}$), the reasoner proceeds to spatial validation. It treats the bounding box $E^*.b$ as a location hypothesis and tests if this specific region in the query image $I$ plausibly contains the target anatomy. This yields a second judgment $v_p \in \{\text{True}, \text{False}\}$.

3. **Step 3: Choose a Policy.** Finally, the framework executes a policy based on the validation outcomes:

   - **Policy 1: Adopt Evidence.** Used if $v_c \wedge v_p$. The framework confirms the retrieved evidence is valid and directly uses $E^*.b$ as the prompt.

- **Policy 2: Concept-guided Search.** Used if $v_c \wedge \neg v_p$. The framework accepts the visual concept but rejects the specific location, using the evidence features to guide a new VLM-driven search in $I$.
- **Policy 3: Zero-shot Reasoning.** Used if $\neg v_c$. The framework rejects the evidence entirely (Tier 3 fallback) and relies on the VLM's internal knowledge.

The VLM generates a structured text output documenting this reasoning chain. This entire process, which leverages the full candidate context for deliberation, is formalized as:

$$B^* = \text{ERA-Reasoner}(I, C, E_{\text{cand}}; \Phi) \tag{2}$$

where $B^*$ is the final spatial prompt and $E_{\text{cand}}$ is the retrieved evidence set. The full algorithm is detailed in Algorithm 1 in the Appendix.

## 4 EXPERIMENTS

### 4.1 EXPERIMENTAL SETUP

**Datasets and Data Integrity**     Our experiments are conducted on a diverse set of medical imaging datasets. We use the ISIC 2018 dataset Codella et al. (2018) for standard scenarios featuring well-defined targets, and tasks from the Medical Segmentation Decathlon (MSD) Simpson et al. (2019) and BraTS 2021 Baid et al. (2021) for complex scenarios characterized by low-contrast targets and intricate anatomical structures. We implemented rigorous measures to ensure a fair evaluation and prevent data leakage across all benchmarks. The full details are provided in Section C.2.

**Baselines and Metrics**     We compare ERA against two baseline categories: (1) **Supervised Specialist Models**, which for 2D tasks include U-Net Ronneberger et al. (2015), ResU-Net, RecU-Net, and R2U-Net Alom et al. (2018), and for 3D tasks include CerebriuDIKU, NVDLMED, Kim et al. Kim et al. (2019), C2FNAS Yu et al. (2020), DINTS He et al. (2021), and nnU-Net Isensee et al. (2019); and (2) **Zero-shot Generalist Models**, which include YOLO-World Cheng et al. (2024), Grounding DINO Liu et al. (2023), FG-CLIP Xie et al. (2025), and MedSAM Ma et al. (2024).

For evaluation, we report Sensitivity (SE), Specificity (SP), F1-Score, Accuracy (AC), and Dice Coefficient (DC) for 2D tasks. For 3D tasks, we use the Dice Similarity Coefficient (DSC), Normalized Surface Distance (NSD). We also report total inference time in seconds for efficiency analysis.

### 4.2 MAIN QUANTITATIVE RESULTS

#### 4.2.1 PERFORMANCE ON STANDARD SCENARIOS

On the ISIC 2018 benchmark (Table 1), ERA demonstrates a strong balance between sensitivity and precision. While the baseline YOLO-World achieves a high DC score (0.9021), its extremely low Specificity (SP) of 0.0817 indicates severe over-segmentation, rendering it clinically unreliable. In stark contrast, our ERA framework achieves a competitive DC of 0.8701 with a near-perfect SP of 0.9851, far outperforming other zero-shot approaches in balanced performance. Notably, ERA is also highly competitive with fully-supervised specialist models, rivaling even the R2U-Net (t=3) configuration.

#### 4.2.2 PERFORMANCE ON COMPLEX SCENARIOS

ERA's superiority is most evident in complex scenarios like the MSD tasks, where generalist models suffer a catastrophic performance collapse with near-zero DSC scores (Table 2). By grounding its reasoning in a medical knowledge base, ERA is the only zero-shot framework to maintain robust, clinically viable performance. Most impressively, on the Prostate dataset, ERA achieves a DSC of 0.8462, outperforming the fully-supervised state-of-the-art nnUNet (0.8311). This result demonstrates that for specialized domains, an evidence-based approach can surpass even models trained extensively on task-specific data.

Table 1: Performance on the ISIC 2018 task. The parameter t indicates the number of unfolding time steps for the recurrent convolutional layers.

| Method | SE↑ | SP↑ | F1↑ | AC↑ | DC↑ |
|---|---|---|---|---|---|
| U-Net (t=2) | 0.9479 | 0.9263 | 0.8682 | 0.9314 | 0.8476 |
| ResU-Net (t=2) | 0.9454 | 0.9338 | 0.8799 | 0.9367 | 0.8567 |
| RecU-Net (t=2) | 0.9334 | 0.9395 | 0.8841 | 0.9380 | 0.8592 |
| R2U-Net (t=2) | 0.9496 | 0.9313 | 0.8823 | 0.9372 | 0.8608 |
| R2U-Net (t=3) | 0.9414 | 0.9425 | 0.8920 | 0.9424 | 0.8616 |
| ERA + SAM2 | 0.8306 | 0.9851 | 0.8701 | 0.9639 | 0.8701 |
| ERA + MedSAM | **0.9657** | **0.9883** | **0.9460** | **0.9852** | **0.9460** |
| YOLO-World + SAM2 | 0.9418 | 0.0817 | 0.8216 | 0.8236 | 0.9021 |
| Grounding DINO + SAM2 | 0.7825 | 0.2595 | 0.1385 | 0.3313 | 0.2433 |
| FG-CLIP + SAM2 | 0.3523 | 0.6621 | 0.3343 | 0.3948 | 0.5011 |
| SAM2 | 0.0258 | **0.9968** | 0.0493 | 0.8634 | 0.0493 |
| ERA + SAM2 | 0.8306 | 0.9851 | 0.8701 | 0.9639 | 0.8701 |
| MedSAM | 0.8679 | 0.1472 | 0.2347 | 0.2436 | 0.2347 |
| ERA + MedSAM | **0.9657** | 0.9883 | **0.9460** | **0.9852** | **0.9460** |

Table 2: Performance comparison on specialized medical segmentation tasks from the MSD.

| Method | Heart | | Hippo. | | Prostate | | Spleen | |
|---|---|---|---|---|---|---|---|---|
| | DSC ↑ | NSD ↑ | DSC ↑ | NSD ↑ | DSC ↑ | NSD ↑ | DSC ↑ | NSD ↑ |
| CerebriuDIKU | 0.8947 | 0.9063 | 0.8900 | 0.9742 | 0.7773 | 0.9631 | 0.9500 | 0.9800 |
| NVDLMED | 0.9246 | 0.9557 | 0.8734 | 0.9633 | 0.7801 | 0.9521 | 0.9601 | 0.9972 |
| Kim et al. | 0.9311 | 0.9644 | 0.8942 | **0.9775** | 0.8083 | 0.9654 | 0.9192 | 0.9483 |
| C2FNAS | 0.9249 | 0.9581 | 0.8867 | 0.9731 | 0.8182 | 0.9696 | 0.9628 | 0.9766 |
| DiNTS | 0.9299 | 0.9635 | 0.8916 | 0.9766 | 0.8231 | 0.9739 | 0.9698 | 0.9983 |
| nnUNet | **0.9330** | **0.9674** | **0.8946** | 0.9766 | 0.8311 | 0.9756 | **0.9743** | **0.9989** |
| ERA + SAM2 | 0.6787 | 0.1508 | 0.5694 | 0.4321 | 0.8462 | 0.6242 | 0.8864 | 0.7103 |
| ERA+MedSAM | 0.8873 | 0.8656 | 0.7948 | 0.9470 | **0.9568** | **0.9976** | 0.9604 | 0.9768 |
| YOLO-World + SAM2 | 0.0366 | 0.1397 | 0.0081 | 0.0776 | 0.0296 | 0.0956 | 0.0119 | 0.0407 |
| Grounding DINO + SAM2 | 0.0262 | 0.5002 | 0.1771 | 0.5160 | 0.0851 | 0.4915 | 0.0585 | 0.4171 |
| FG-CLIP + SAM2 | 0.0333 | 0.4799 | 0.1821 | 0.4923 | 0.0913 | 0.4820 | 0.0150 | 0.1428 |
| SAM2 | 0.0031 | 0.0772 | 0.0000 | 0.0051 | 0.0128 | 0.0654 | 0.0010 | 0.0066 |
| ERA + SAM2 | 0.6787 | 0.1508 | 0.5694 | 0.4321 | 0.8462 | 0.6242 | 0.8864 | 0.7103 |
| MedSAM | 0.0137 | 0.0012 | 0.1535 | 0.1212 | 0.0704 | 0.0474 | 0.0254 | 0.0527 |
| ERA + MedSAM | **0.8873** | **0.8656** | **0.7948** | **0.9470** | **0.9568** | **0.9976** | **0.9604** | **0.9768** |

## 4.3 EFFICIENCY ANALYSIS

While performance is critical, practical deployment also hinges on computational efficiency. This section analyzes the inference time of the ERA framework as a necessary trade-off for its superior accuracy and reliability.

As detailed in Table 3, the ERA framework's inference time is considerably higher than that of the zero-shot baselines. For instance, on the ISIC 2018 dataset, ERA requires 2151.29 seconds, whereas YOLO-World and Grounding DINO complete in 104.92 and 155.24 seconds, respectively. However, this comparison must be contextualized by performance. The baseline methods, despite their speed, produce clinically unusable results on all specialized tasks, as evidenced by their near-zero DSC scores in Table 2. Their speed, therefore, represents the efficiency of arriving at a wrong answer.

The computational cost of ERA is a deliberate trade-off, investing time in a rigorous retrieval and reasoning process to achieve a massive leap in performance—from complete failure to robust, state-of-the-art results. This investment transforms the paradigm from an unreliable tool into a viable clinical instrument, justifying the additional computational budget.

Table 3: Comparison of total inference time in seconds between the ERA framework and other zero-shot baseline methods across the ISIC 2018 and four MSD datasets.

| Method | | ISIC 2018 | MSD Datasets | | | |
|---|---|---|---|---|---|---|
| | | time(s) | Heart | Hippocampus | Prostate | Spleen |
| Baselines | YOLO-World + SAM2 | 104.92 | 163.13 | 636.72 | 43.59 | 274.72 |
| | Grounding DINO + SAM2 | 155.24 | 601.78 | 2901.46 | 159.82 | 992.26 |
| | FG-CLIP + SAM2 | 138.77 | 254.88 | 792.51 | 67.36 | 432.46 |
| Ours | ERA + SAM2 | **2151.29** | **3309.39** | **11545.83** | **643.39** | **4060.01** |

Table 4: Ablation study of the ERA framework, evaluating performance across all datasets.

| Configuration | ISIC 2018 | | | Heart | | Hippo. | | Prostate | | Spleen | | BraTS 2021 | |
|---|---|---|---|---|---|---|---|---|---|---|---|---|---|
| | SE↑ | SP↑ | DC↑ | DSC↑ | NSD↑ | DSC↑ | NSD↑ | DSC↑ | NSD↑ | DSC↑ | NSD↑ | Dice↑ | mIoU↑ |
| *Ablations* | | | | | | | | | | | | | |
| w/o Reasoning | 0.60 | 0.91 | 0.55 | 0.67 | 0.14 | 0.49 | 0.40 | 0.79 | 0.52 | 0.87 | 0.64 | 0.76 | 0.65 |
| w/o Retrieval | **0.83** | 0.87 | 0.84 | 0.06 | 0.00 | 0.14 | 0.25 | 0.07 | 0.03 | 0.04 | 0.05 | 0.27 | 0.17 |
| w/o Tier-2 | 0.57 | 0.89 | 0.51 | 0.57 | 0.13 | 0.50 | 0.41 | 0.80 | 0.56 | 0.76 | 0.68 | **0.78** | **0.66** |
| Unguided SAM2 | 0.03 | **1.00** | 0.05 | 0.00 | 0.08 | 0.00 | 0.01 | 0.01 | 0.07 | 0.00 | 0.01 | 0.02 | 0.01 |
| **ERA + SAM2** | **0.83** | 0.99 | **0.87** | **0.68** | **0.15** | **0.57** | **0.43** | **0.85** | **0.62** | **0.89** | **0.71** | **0.78** | **0.66** |

## 4.4 ABLATION STUDIES

Our ablation studies, detailed in Table 4 and Table 5, reveal an indispensable synergy between evidence retrieval and deliberative reasoning that enhances both performance and efficiency. Ablating either component causes a severe performance collapse. For instance, without the retrieval module (w/o Retrieval), the VLM's implicit knowledge is insufficient, causing the Heart DSC to plummet from 0.68 to 0.06. Conversely, removing the reasoning module (w/o Reasoning) leads to a significant degradation, with the Spleen DSC dropping from 0.89 to 0.76, demonstrating that evidence alone is not enough without structured interpretation. Counterintuitively, the retrieval module also acts as a powerful efficiency booster. While reasoning contributes to inference time, the w/o Retrieval configuration is by far the most computationally expensive, taking nearly 7000 seconds on ISIC 2018. This shows that retrieval, by providing focused evidence, critically prunes the search space, making subsequent deliberation far more efficient than an unguided, brute-force approach. The complete ERA framework thus strikes an optimal balance, where both components work in concert to maximize performance and computational feasibility.

## 4.5 QUANTITATIVE ANALYSIS OF TIER EFFECTIVENESS

**Inferred Tier Contribution** To strictly quantify the distinct necessity of each reasoning tier, we introduce the Relative Performance Drop (RPD) metric, defined as the percentage decline in Dice score when a specific tiering capability is ablated. Table 6 presents this analysis across all datasets, including an aggregated average to highlight overall systemic dependencies.

First, Tier 3 (zero-shot fallback) proves functionally insufficient for specialized medical domains, with a substantial average RPD of 70.3%. However, this dependency exhibits significant variance: the RPD is negligible for ISIC (3.4%) but exceeds 90% for MSD tasks. This disparity highlights the critical role of the domain gap: while skin lesions share visual features with natural images found in VLM pre-training (rendering zero-shot feasible), the specialized, low-contrast anatomy of cross-sectional organ imaging is entirely alien to the model, strictly necessitating the external evidence provided by the retrieval module.

Second, Tier 1 (direct adoption) establishes a robust foundational baseline, reflected in a moderate average RPD of 10.7% when reasoning is removed. Notably, for structurally consistent targets like the Heart and Spleen, the RPD is minimal (1.5% and 2.2%, respectively). This indicates that for such "simple" scenarios characterized by stable morphology, the retrieved evidence is sufficiently precise

Table 5: Ablation study of inference time in seconds for the ERA framework and its different configurations.

| Configuration | ISIC 2018 | Heart | Hippocampus | Prostate | Spleen | BraTS 2021 |
|---|---|---|---|---|---|---|
| w/o Reasoning | 2148.66 | 3316.50 | 11575.59 | 643.35 | 3914.52 | 2487.23 |
| w/o Retrieval | **6987.60** | **10554.95** | **36510.91** | **2014.39** | **12181.06** | **7866.00** |
| w/o Tier-2 | 2079.83 | 3206.49 | 11167.77 | 652.73 | 3770.90 | 2407.64 |
| Unguided SAM2 | 55.01 | 93.21 | 369.94 | 25.06 | 163.50 | 80.65 |
| **ERA + SAM2** | 2151.29 | 3309.39 | 11545.83 | 643.39 | 4060.01 | 2527.86 |

for direct mapping. The framework's stability here validates its design efficiency: it correctly filters these clear-cut cases via Tier 1, reserving the computational cost of deliberation for more ambiguous targets.

Finally, Tier 2 (deliberative reasoning) functions as the critical refinement engine for complex scenarios. In contrast to the simple cases above, removing Tier 2 causes sharp performance penalties in challenging tasks, such as a 14.0% drop in Hippocampus and a 36.8% drop in ISIC segmentation. These elevated RPD values confirm that for intricate or variable targets, the naive adoption of evidence is inadequate, and the deliberative reasoning process becomes indispensable for error correction.

Table 6: Inferred contribution of reasoning tiers based on Relative Performance Drop (RPD). The Average row highlights the system's overall reliance on each tier. The data confirms a dynamic dependency: Tier 1 suffices for morphologically consistent anatomy (Heart, Spleen), while Tier 2 is essential for complex tasks (Hippocampus, ISIC).

| Dataset | Full | w/o Retrieval (Tier 3 Only) | | w/o Reasoning (No Tier 2) | |
|---|---|---|---|---|---|
| | Score | Score | RPD ($\Delta\%$) | Score | RPD ($\Delta\%$) |
| ISIC 2018 | 0.87 | 0.84 | 3.4% | 0.55 | 36.8% |
| Heart | 0.68 | 0.06 | 91.2% | 0.67 | 1.5% |
| Hippocampus | 0.57 | 0.14 | 75.4% | 0.49 | 14.0% |
| Prostate | 0.85 | 0.07 | 91.8% | 0.79 | 7.1% |
| Spleen | 0.89 | 0.05 | 94.4% | 0.87 | 2.2% |
| BraTS 2021 | 0.78 | 0.27 | 65.4% | 0.76 | 2.6% |
| **Average** | **-** | **-** | **70.3%** | **-** | **10.7%** |

### 4.6 AUDITABLE WORKFLOW FOR DECISION TRACEABILITY

To address the inherent opacity of VLMs in medical applications, ERA is designed to structure the decision process into an auditable reasoning chain. Specifically, the framework documents the intermediate steps: the retrieved visual evidence, the sequential validation logic, and the final policy adoption (details in Appendix Figure 4). Crucially, this workflow renders the decision logic explicit, allowing for easier identification of potential failures, such as cases where retrieved evidence is rejected. While establishing genuine clinical trust requires extensive validation with medical professionals, we believe the evidence-based traceability offered by ERA provides the necessary audit trail to support such future assessments.

### 4.7 QUALITATIVE ANALYSIS AND DISCUSSION

**Qualitative Analysis**   As shown in Figure 3, our ERA framework generates accurate, anatomically plausible segmentations on challenging tasks where baseline models catastrophically fail, producing unstructured noise or incorrect shapes (see Appendix D for a detailed analysis).

**Discussion**   Our results show that ERA performs well in medical imaging because it changes the core process from simple pattern matching to explicit, evidence-based reasoning. By grounding its

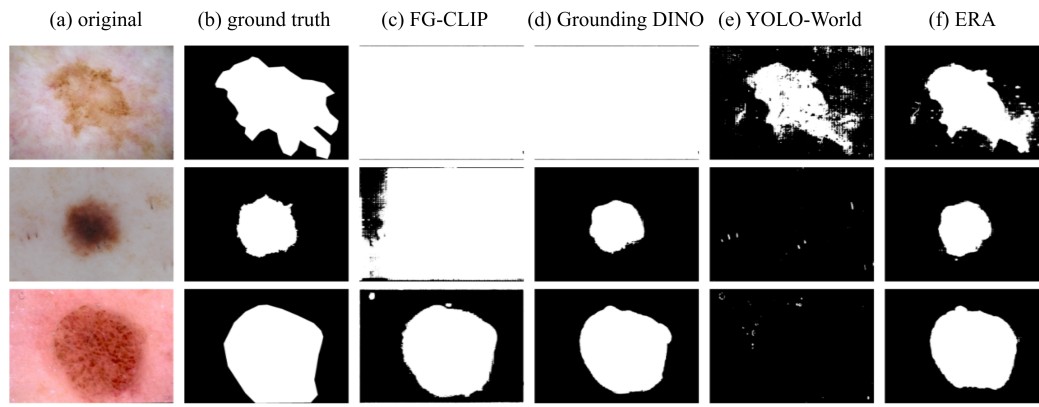

Figure 3: Qualitative comparison on challenging examples from the ISIC 2018. Further visualizations can be found in the appendix.

decisions in an external knowledge base, ERA avoids the internal biases of VLMs. This is why it remains robust on complex tasks like the MSD challenges, where other generalist models that rely on flawed internal knowledge fail completely. The framework's main strength comes from combining RAG, which provides the necessary evidence, with CoT, which ensures that evidence is used in a careful and logical way.

The primary limitation of ERA is its slow inference speed, a common problem for large VLMs. This highlights a key trade-off in the field: the powerful, large-scale models needed for complex reasoning are computationally expensive. This makes speed a critical area for future work. Research could focus on model compression, knowledge distillation, or creating more efficient reasoning methods to make evidence-based frameworks like ERA practical for real-time clinical use. ERA's ability to create a transparent and reviewable reasoning path offers a vital step toward building the trust required to integrate advanced AI into high-stakes medical workflows.

## 5 CONCLUSION

Large Vision-Language Models often fail in medical imaging because they rely on opaque, internal knowledge, limiting their reliability and auditability for clinical use. To solve this, we developed ERA, a zero-shot framework that mitigates this by grounding VLM reasoning in an external, verifiable knowledge base. By combining RAG to source evidence with a CoT process to ensure its logical use, ERA transforms the inference process from implicit guessing to explicit, evidence-based inference. Experiments show this training-free approach not only remains robust in complex scenarios where others fail but can also match or exceed fully-supervised specialist models. By generating a transparent and traceable reasoning path, ERA offers a verifiable workflow that facilitates human oversight for medical AI. While computational efficiency remains a challenge, our work presents a crucial step toward building the more accountable and reliable AI systems required for high-stakes clinical applications.

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

## APPENDIX

This supplementary document provides additional details, analyses, and visualizations to support our main paper.

- **Section A** clarifies that Large Language Models were used exclusively for polishing the manuscript's text to improve readability and did not contribute to any core scientific content or results.

- **Section B** provides a comprehensive guide to the framework's implementation for full reproducibility. This includes the formal pseudocode for the inference pipeline, a detailed table of all key hyperparameters, visualizations of the prompt templates used in the tiered reasoning engine, and specifics of the retrieval strategy.

- **Section C** details the construction of the medical knowledge base, including the data sources from MedIMeta, the curation process, and the critical measures taken to ensure data integrity and prevent leakage during evaluation. It also confirms the availability of the source code.

- **Section D** presents an in-depth qualitative analysis, supplementing the main paper with additional visualizations that highlight the ERA framework's robust performance in contrast to the catastrophic failures of baseline models on complex tasks.

- **Section E** delivers a detailed quantitative breakdown of the framework's detection performance, presenting comprehensive metrics in tables that compare ERA against all baselines across the ISIC, BraTS, and MSD datasets.

## A    STATEMENT ON THE USE OF LARGE LANGUAGE MODELS

To enhance the readability and reduce grammatical errors in this paper, we utilized a Large Language Model (LLM) for the sole purpose of polishing the manuscript's text. The scope of its use was strictly confined to refining language and improving clarity. The LLM was not involved in generating the core content, formulating the research ideas, conducting the experiments, or analyzing the results. All intellectual contributions and scientific claims presented herein are the original work of the authors.

## B    IMPLEMENTATION DETAILS AND REPRODUCIBILITY

This section provides key implementation details to ensure reproducibility, addressing hyperparameter settings, the reasoning mechanism, and the retrieval strategy.

### B.1    ALGORITHM

The complete ERA inference pipeline is formally detailed in Algorithm 1 below.

### B.2    FRAMEWORK AND HYPERPARAMETER SETTINGS

Key hyperparameters for the ERA framework are provided in Table 7. For baseline models, we used their official pre-trained weights and default inference settings. The logic thresholds are presented as effective ranges, with the optimal value determined on a validation set for each domain.

---

**Algorithm 1** The ERA Framework Inference Pipeline

---

1: **Input:** Query Image $I$, Natural Language Instruction $C$
2: **Parameters:** Knowledge Base $\mathcal{K}$, VLM Reasoner $\Phi$, Thresholds $T_{high}$
3: **Output:** High-Confidence Spatial Prompt $B^*$

4: **function** ERA-INFERENCE($I, C$)
5:     $E_{cand} \leftarrow$ Retrieve($I, C; \mathcal{K}, k$)                           ▷ Retrieve $k$ candidates
6:     **if** $E_{cand} = \emptyset$ **then**
7:         **return** $\Phi_{ZS}(I, C)$                            ▷ Tier 3: Fallback
8:     **end if**
9:     $E^* \leftarrow E_{cand}[0]$                              ▷ Top-1 candidate
10:     $s^* \leftarrow$ Similarity($E^*, I$)                  ▷ Calculate similarity score

11:     **if** $s^* > T_{high}$ **then**                     ▷ Tier 1: High Confidence
12:         $B^* \leftarrow E^*.b$                 ▷ Directly adopt Top-1 evidence
13:     **else**                   ▷ Tier 2: Ambiguous, need deliberation
14:                   ▷ VLM reasons over the full candidate set $E_{cand}$
15:         $v_c, v_p \leftarrow \Phi_{Deliberate}(I, C, E_{cand})$
16:         **if** $v_c \wedge v_p$ **then**         ▷ Policy 1: Adopt Evidence (after validation)
17:             $B^* \leftarrow E^*.b$
18:         **else if** $v_c$ **then**             ▷ Policy 2: Concept-guided Search
19:             $B^* \leftarrow \Phi_{Search}(I, C, E_{cand})$
20:         **else**               ▷ Policy 3: Zero-shot Reasoning
21:             $B^* \leftarrow \Phi_{ZS}(I, C)$
22:         **end if**
23:     **end if**
24:     **return** $B^*$
25: **end function**

---

Table 7: Key hyperparameters for the ERA Framework.

| Category | Parameter | Value / Description |
|---|---|---|
| Retrieval | top_k | 6 |
| | image_text_weight | 0.95 |
| Reasoning Logic | tier1_similarity | Range: [0.93 − 0.96] |
| | tier2_similarity | Range: [0.82 − 0.88] |
| LMM Engine (Qwen) | Model | Qwen2.5-VL-7B-Instruct |
| | Quantization | 8-bit |
| | Attention Mechanism | Standard Eager Attention |
| | Dtype | torch.bfloat16 |
| Segmentation (SAM2) | Model | SAM2 with Hiera-B+ Image Encoder |
| | Checkpoint | sam2.1_hiera_base_plus.pt |

### B.3 TIERED REASONING AND PROMPT TEMPLATES

Our framework's tiered reasoning mechanism, illustrated in Figure 2 of the main paper, is detailed in Figure 4. This diagram provides a comprehensive visualization of the process, detailing the specific prompt template used at each stage of the decision-making flow to ensure full reproducibility.

### B.4 RETRIEVAL STRATEGY DETAILS

The candidate selection mechanism, denoted as get_best_candidate, is implemented through a multi-stage retrieval and ranking process. Initially, the retriever module evaluates all candidates from the knowledge base, assigning each a composite final score that combines both content and size similarity. The module subsequently returns a ranked list of the top-k candidates, where k is set

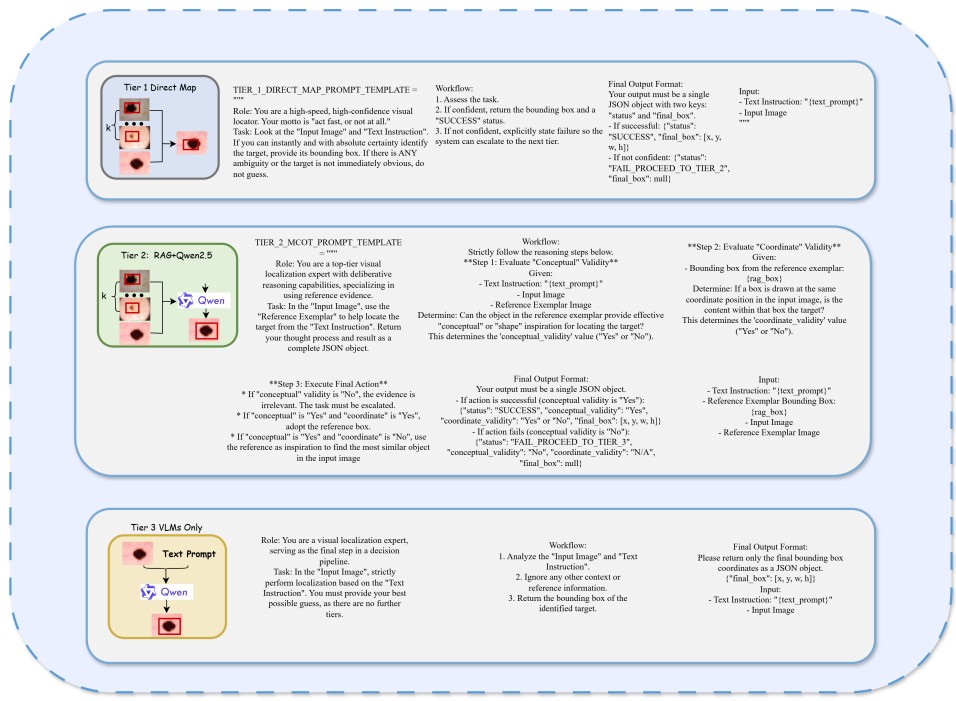

Figure 4: Detailed visualization of the three-tiered reasoning mechanism. Each tier—(1) Direct Map, (2) RAG with VLM, and (3) VLM-Only Fallback—is governed by a specific prompt template that dictates the model's behavior and decision criteria. As shown in the figure, the complete prompt template for each tier is displayed, which includes a role definition , task description , workflow , and specifications for the input/output format.

to 6 in our experiments. The get_best_candidate operation then formally selects the highest-ranked candidate from this list. This top-ranked candidate serves as the primary evidence, $E^*$, for the deliberative reasoning module.

# C  KNOWLEDGE BASE CONSTRUCTION AND DATA USAGE

## C.1  DATA INTEGRITY AND LEAKAGE PREVENTION

To ensure a fair evaluation and prevent data leakage, our knowledge base was built exclusively from the training splits of source datasets, with all test benchmark data strictly excluded. Furthermore, an inference-time filter prevents a query from retrieving itself, guaranteeing that performance relies on genuine knowledge transfer rather than data leakage.

## C.2  KNOWLEDGE BASE COMPOSITION AND CONSTRUCTION

To support our evidence-based reasoning framework, we constructed a large-scale, diverse medical visual knowledge base. The data for this knowledge base was sourced from MedIMeta Woerner et al. (2025), a large, standardized, multi-domain meta-dataset containing high-quality medical images with ground-truth annotations from 10 different medical domains, including dermatoscopy, CT, and X-ray.

Our construction process programmatically curated these source datasets into a unified knowledge base. For each source image with a corresponding ground-truth segmentation mask, we computed a precise bounding box to serve as the geometric anchor. This process resulted in a final JSON manifest where each entry consistently links an image path to a predefined text label and its corresponding bounding box coordinates. The manifest was then used to build a feature matrix by encoding each image into a normalized feature vector using a pre-trained BLIP model.

### C.3 CODE AVAILABILITY

To facilitate further research and ensure full reproducibility, our code is included in the supplementary material provided with this submission.

## D DETAILED QUALITATIVE ANALYSIS

To supplement the brief analysis in the main paper, this section provides a more in-depth discussion of our qualitative results with visualizations in Figure 3 and Figure 5. While the ERA framework demonstrates strong performance on 2D tasks like ISIC 2018 by producing coherent and well-defined boundaries, its superiority becomes most evident in highly specialized and demanding tasks. In these scenarios, such as MSD organ and BraTS tumor segmentation, baseline models exhibit catastrophic failures, frequently degenerating into geometrically incorrect shapes, fragmented predictions, or unstructured noise that bears little resemblance to the target anatomy. In striking contrast, our ERA framework consistently reconstructs the correct anatomical structures with high fidelity, accurately delineating organ boundaries in MSD while respecting their 3D topology, and precisely identifying tumor sub-regions in BraTS. These results visually confirm that ERA's evidence-based reasoning paradigm enables it to effectively adapt its knowledge to diverse and highly specialized clinical scenarios where generalist approaches fall short.

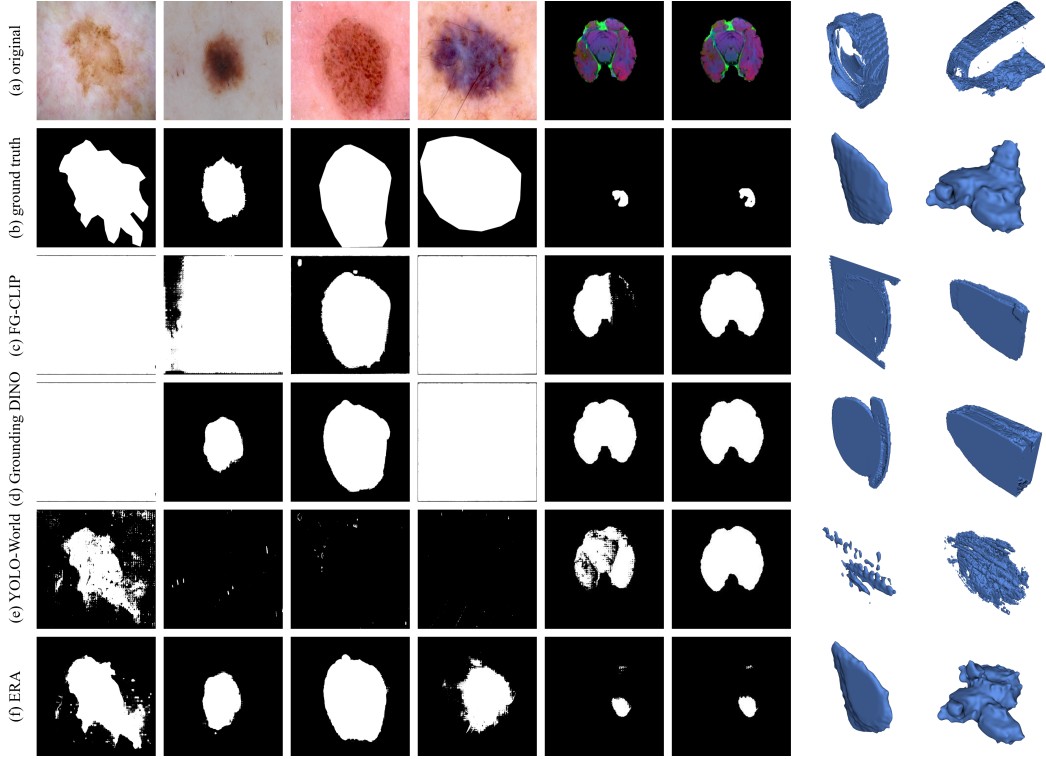

Figure 5: Additional qualitative examples from the MSD and BraTS datasets. This figure provides more extensive visualizations, showcasing ERA's consistent performance on a wider range of challenging 3D medical imaging cases compared to the noisy and inaccurate results from baseline models.

## E DETAILED QUANTITATIVE PERFORMANCE

This section provides a detailed quantitative breakdown of the open-vocabulary detection performance. Table 8 presents a consolidated comparison across all six evaluated datasets, including ISIC

Table 8: Comprehensive performance comparison of the ERA framework against baselines across six medical datasets.

| ISIC 2018 | | | | | BraTS | | | | |
|---|---|---|---|---|---|---|---|---|---|
| **Model** | Recall | Prec. | F1 | mIoU | **Model** | Recall | Prec. | F1 | mIoU |
| FGCLIP | 0.9518 | 0.9518 | 0.9518 | 0.3807 | FGCLIP | 0.3914 | 0.4705 | 0.4273 | 0.1247 |
| GroundingDINO | 0.9833 | 0.5869 | 0.7351 | 0.7044 | GroundingDINO | 0.6002 | 0.5115 | 0.5523 | 0.1519 |
| YOLOWORLD | 0.9418 | 0.7286 | 0.8216 | 0.8217 | YOLOWORLD | **1.0000** | 0.0968 | 0.1765 | 0.0968 |
| ERA | **1.0000** | **1.0000** | **1.0000** | **0.8424** | ERA | 0.8174 | **0.8034** | **0.8103** | **0.5788** |

| MSD: Heart | | | | | MSD: Hippocampus | | | | |
|---|---|---|---|---|---|---|---|---|---|
| **Model** | Recall | Prec. | F1 | mIoU | **Model** | Recall | Prec. | F1 | mIoU |
| FGCLIP | 0.5697 | 0.3743 | 0.4518 | 0.1065 | FGCLIP | **0.9474** | 0.6838 | 0.7943 | **0.4477** |
| GroundingDINO | **0.9035** | 0.9629 | 0.9322 | **0.7224** | GroundingDINO | 0.8691 | **0.9895** | 0.9254 | 0.5155 |
| YOLOWORLD | 0.0355 | 0.0191 | 0.0248 | 0.0186 | YOLOWORLD | 0.0081 | 0.0041 | 0.0054 | 0.0041 |
| ERA | 0.9028 | **0.9978** | **0.9479** | 0.6708 | ERA | 0.9463 | 0.9610 | **0.9536** | 0.4087 |

| MSD: Prostate | | | | | MSD: Spleen | | | | |
|---|---|---|---|---|---|---|---|---|---|
| **Model** | Recall | Prec. | F1 | mIoU | **Model** | Recall | Prec. | F1 | mIoU |
| FGCLIP | 0.8973 | 0.7110 | 0.7933 | 0.1785 | FGCLIP | 0.5524 | 0.1603 | 0.2485 | 0.2347 |
| GroundingDINO | **0.9958** | 0.7890 | 0.8804 | **0.8704** | GroundingDINO | 0.9896 | 0.2406 | 0.3871 | **0.8952** |
| YOLOWORLD | 0.0291 | 0.0153 | 0.0201 | 0.0150 | YOLOWORLD | 0.0118 | 0.0062 | 0.0081 | 0.0060 |
| ERA | **0.9958** | **1.0000** | **0.9979** | 0.8382 | ERA | **0.9915** | **1.0000** | **0.9957** | 0.8690 |

2018, BraTS, and four tasks from the MSD. To rigorously assess both the localization accuracy and detection completeness, we employ a standard set of metrics anchored on Intersection over Union (IoU). Specifically, we report Recall (sensitivity), defined as the ratio of correctly matched ground truths to the total number of ground truths; Precision, the ratio of correct matches to the total number of predicted boxes; and F1-Score, the harmonic mean of Recall and Precision. Furthermore, to evaluate the geometric quality of the detected regions, mIoU is calculated specifically on the True Positive matches.

