# OpenReview forum: "ERA: Evidence-Based Reasoning and Augmentation for Open-Vocabulary Medical Vision"
_ICLR.cc/2026/Conference — ICLR 2026 Conference Withdrawn Submission_

### Official Review · Reviewer_ugzA · 2025-10-27

**Soundness:** 2
**Presentation:** 2
**Contribution:** 2
**Rating:** 2
**Confidence:** 4

**Summary:**

This paper presents ERA, a novel framework addressing the critical need for reliable, evidence-based reasoning in medical VLMs. The proposed training-free approach, which synergizes Retrieval-Augmented Generation (RAG) and Chain-of-Thought (CoT), is well-motivated and technically sound. The framework demonstrates impressive robustness, achieving state-of-the-art zero-shot performance and even outperforming supervised specialists on complex tasks where other generalist models fail completely. However, this strong performance comes at the cost of extremely high inference latency, which is orders of magnitude slower than baselines. While the results are strong, the practical viability is limited by this efficiency trade-off.

**Strengths:**

- **Important Problem:** The work tackles a critical and timely limitation of modern VLMs: their lack of reliability and interpretability, which is a major barrier to adoption in high-stakes domains like medicine.
- **Novel Methodology:** The framework’s design, which combines RAG and CoT to enforce explicit, evidence-based reasoning in a *training-free* manner, is a novel and elegant approach. It smartly redesigns the inference process rather than relying on costly model fine-tuning.
- **Exceptional Robustness:** The empirical results are strong, particularly in complex scenarios. ERA maintains robust performance on MSD tasks where powerful zero-shot baselines (e.g., YOLO-World, Grounding DINO) suffer a "catastrophic performance collapse" with near-zero scores.

**Weaknesses:**

- **Computational Cost:** The most significant weakness is the inference efficiency. As shown in Table 3 and acknowledged in Section 4.3, ERA's inference time is orders of magnitude higher than the baselines. This severe latency is a major barrier to practical clinical deployment.
- **Knowledge Base Dependency:** The framework's effectiveness is fundamentally dependent on the quality and comprehensiveness of the external knowledge base (K). The main paper provides limited detail on its scale and composition (deferring to Appendix C). It is unclear how the model performs on "out-of-distribution" queries that lack close exemplars in the knowledge base.
- **Unclear Baseline Configuration:** In Table 1 and Table 2, results are shown for both "ERA + SAM2" and "ERA + MedSAM". The baseline "MedSAM"  scores exceptionally poorly in Table 1, which is highly counter-intuitive for a model specialized in medical segmentation. This discrepancy is not explained and makes it difficult to assess the true contribution of ERA when paired with MedSAM.
- **Missing Ablation of Reasoning Policies:** The Deliberative Reasoning Engine (Sec 3.3.2) uses a crucial three-policy CoT (Adopt, Search, Zero-shot). However, the paper does not include an ablation study on the impact of these policies. Analyzing the activation frequency of each policy or the performance impact of removing one (e.g., always adopting evidence) would significantly strengthen the validation of this core component. No direct evidence found in the manuscript.

**Questions:**

See Weakness.

**Details Of Ethics Concerns:**

No Ethics Concerns.

---

> ### Author Response · Authors · 2025-11-14
> **Response to Weaknesses and Questions**
>
> We are very grateful for the valuable time you spent reviewing our work and for your insightful and constructive feedback. We will now respond to your concerns point by point.
>
> ### 1. On Missing Ablation of Reasoning Policies
>
> We appreciate your close attention to our core component, the Deliberative Reasoning Engine (Sec 3.3.2). However, we believe the claim that "No direct evidence found" is something worth reconsidering. We argue that our ablation study in Table 4 already provides the direct evidence you are asking for, by demonstrating the necessity of the complete three-policy CoT through component removal.
>
> 1. **Impact of "always adopting evidence"**: This scenario is precisely modeled by our **w/o Reasoning** configuration in Table 4. In this setup, the CoT validation (Sec 3.3.2) is removed, forcing the model to blindly adopt the retrieved evidence (always using Policy 1). The results show a severe performance degradation (Spleen DSC drops from 0.89 to 0.76), proving that the reasoning and validation steps are essential.
> 2. **Impact of the policies (Policy 3)**: This is modeled by our **w/o Retrieval** configuration in Table 4. When retrieval is removed, the framework is forced to *only* use Policy 3 (Zero-shot Reasoning). The results show a catastrophic performance collapse (Heart DSC plummets from 0.68 to 0.06).
>
> In addition, another reviewer also inquired about our ablation study: 'For table 4, what is the difference of "w/o Retrieval" and "w/o Tier2"? My understanding is w/o Retrieval means NO retrieval at all (Tier 3 only). The ablation w/o Reasoning means retrieval happens, but only the simple Tier 1 logic is used.' This directly proves that we did, in fact, include an ablation study.
>
> ### 2. On Computational Cost
>
> We fully agree that computational cost is a significant concern. As we stated in Section 4.3, this is a deliberate trade-off.
>
> The baseline methods, while fast, suffer a catastrophic performance collapse on complex medical tasks like MSD, as shown in Table 2, with DSC scores approaching zero. As we discuss in Section 4.3, the speed of these baselines merely represents the efficiency of arriving at a wrong answer.
>
> The contribution of ERA is its ability to achieve robust, zero-shot performance that is comparable to, and in some cases even exceeds, fully-supervised specialist models. For instance, as shown in Table 2, on the Prostate dataset, our zero-shot ERA achieves a DSC of 0.8462, outperforming the fully-supervised state-of-the-art nnUNet (0.8311). We acknowledge that the high latency, due to using a VLM as the reasoning engine, is a primary limitation, as discussed in Section 4.6. Our future work will focus on efficiency optimization, such as knowledge distillation, as mentioned in our discussion.
>
> ### 3. On Knowledge Base Dependency
>
> Your observation regarding the dependency on the knowledge base (K) is correct; its quality is fundamental to the framework's effectiveness. We quantified this in our ablation study.
>
> As shown in Table 4, the w/o Retrieval configuration demonstrates this dependency, with performance collapsing (e.g., Heart DSC drops from 0.68 to 0.06) when the knowledge base is removed.
>
> Regarding performance on "out-of-distribution" queries (where exemplars are lacking), the paper has designed explicit mechanisms to handle this.
>
> 1. As described in Section 3.3.2, our CoT includes Policy 3: Zero-shot Reasoning. When retrieved evidence is deemed conceptually mismatched ($\neg v_c$), ERA ignores the evidence and reverts to the VLM's intrinsic zero-shot capabilities.
> 2. Furthermore, as detailed in Algorithm 1 in the Appendix, if the knowledge base returns no candidates ($E_{cand}=\emptyset$), the framework automatically executes $\Phi_{ZS}(I,C)$, which is the pure zero-shot inference path.
>
> These two mechanisms ensure that the framework remains robust even when faced with OOD queries.

---

> ### Author Response · Authors · 2025-11-14
> **Supplement to 'Response to Weaknesses and Questions'**
>
> ### 4. On Unclear Baseline Configuration
>
> We thank you for pointing out this ambiguity, and we agree with your intuition that the low scores for MedSAM are counter-intuitive. We apologize for the lack of clarity and will revise this in the paper.
>
> As we discussed in Section 2.1, the performance of prompt-based models like MedSAM is fundamentally contingent on the quality of the input prompts they receive.
>
> The MedSAM (Baseline) in Table 1 and Table 2 represents the model's unguided performance without any effective spatial prompts. We designed this experiment to simulate a fully automated workflow where manual prompting is not feasible. This is further supported by the Unguided SAM2 ablation in Table 4, which also shows near-zero performance.
>
> The catastrophic failure of the unguided baseline (DC 0.2347 in Table 1) highlights our core motivation: high-quality spatial prompts are essential. When ERA is combined with MedSAM (ERA + MedSAM), the performance dramatically increases from 0.2347 to 0.9460. This strongly demonstrates the value of ERA as a high-quality, automatic prompt generator.
>
> Thank you again for your feedback. We hope our response has addressed your concerns, and we look forward to further discussion with you.

---

### Official Review · Reviewer_GRgY · 2025-10-31

**Soundness:** 3
**Presentation:** 2
**Contribution:** 3
**Rating:** 4
**Confidence:** 4

**Summary:**

This work leverages an external database from MedIMeta to construct a knowledge base for an RAG system in medical tasks. The image, text and corresponding bounding box annotations are retrieved and a multi-stage prompting strategy is designed to decide whether or not to leverage this retrieved image/bounding box locations in the prompt of another multimodal large language model. Experiments are conducted in multiple datasets to evaluate the performance.

**Strengths:**

- This work explicitly incorporates the evidence as its reasoning base in a multimodal large language model’s setting.
- A multi-stage prompting strategy is designed to better incorporate the retrieved information.
- It’s interesting to retrieve spatial annotations with the images to improve the localization capability.

**Weaknesses:**

*Contribution*
-  The contribution is overclaimed. [line 82] It’s at most encouraging the usage, instead of forcing it to check evidence.

-	[Line 96] it’s hard to say this framework “transforms the decision process into a transparent process”. Any reasoning model can achieve this.

-	[line 53] Claiming “CoT is traceable” is too strong. It at most offers some insights. Whether the generated chains are correct requires expert review and whether CoT is the real trace is also arguable.

*Implementation*

Many implementation details are missing
- [Line 187] what is a spatial prompt? What text and what image is retrieved?
- [Line 222] how exactly is a a query made of an image and a text encoded? What is the text in the knowledge base /where do they come from? Simply class name?
- [Line 209] many medical images are not annotated with bounding box or segmentation masks. Are you retrieving the whole image or only the cropped part?
- How do you handle resolution, ratio difference between retrieved and tested images?

*Method and Data*

- [Table 8] it seems directly applying groundingDINO is better than the proposed approach? Since this framework is training-free, can this approach be combined with groundingDINO? If not, why do people not directly apply GroundingDINO?

-	[line 21] It’s unclear whether the compared specialist model is trained on the same external database used to construct RAG. If not, the comparison is unfair.

-	Is there any data leackage checked? Does the database potentially include the test set datasets?

**Questions:**

- [line 252] Why is it necessary to check the ground-truth bounding box of retrievd images again?

- [Line 259] Under what scenario will it happen that a bounding box from the knowledge base’s image is directly useable in the tested images? This design seems somewhat unreasonable to me.

- How is SAM2 used in this framework? Usef to offer the bounding box of the tested image or the retrived image?

- [line 771] Why is it necessary to compute another bounding box as “geometric anchor” when ground-truth mask is already present?

---

> ### Author Response · Authors · 2025-11-13
> **Response to Weaknesses and Questions**
>
> We are very grateful for the valuable time you spent reviewing our work and for providing such insightful and constructive feedback. Your in-depth analysis is crucial for us to improve the paper. We will now respond to your concerns point by point.
>
> ### On Contribution
>
> We agree that "forcing" is a strong word. Our aim is not to achieve absolute enforcement, but to provide an auditable, evidence-based reasoning path. Our framework ensures this through a hypothesis-testing path (Section 3.3.2). The model must first evaluate the retrieved evidence $E^*$ and then execute one of the three prescribed policies based on the evaluation results $v_c$ and $v_p$. This structured process ensures that the decision is based on evidence, not arbitrary guessing. Our core contribution lies in transforming this decision process into an auditable workflow (Section 4.5). As stated in Section 3.3.2, the VLM generates a structured text output that records the full reasoning chain: including what evidence was retrieved, how it was verified, and which policy was ultimately adopted. This provides the necessary foundation for clinical review and human-machine collaboration.
>
> ### On Implementation Details
>
> **1. Definition of Spatial Prompt:**
> In our work, Spatial Prompt refers to the bounding boxes used to drive the downstream segmentation model. As we stated in Section 1, our goal is to address the problem of automatically "generate prompts like bounding boxes" and associate it with the automatic creation of spatial prompts.
>
> **2. Query and Knowledge Base Text:**
> The query consists of the input image $I$ and the text instruction $C$. As stated in Section 3.2.2 (Lines 221-222): "During use, a query made of an image I and a text instruction C is encoded into a normalized query vector."
> The text $t$ in the knowledge base is a "predefined text label" that describes the target entity. As stated in Section 3.2.1: "t is a text label that describes the target...". This is also confirmed in Section C.2: "...links an image path to a predefined text label..."
>
> **3. Annotation and Retrieval:**
> The reviewer's observation is very accurate; our framework does not rely on bounding box annotations.
> As stated in Section C.2: "For each source image with a corresponding ground-truth segmentation mask, we computed a precise bounding box to serve as the geometric anchor." We utilize existing ground-truth segmentation masks and compute their minimal bounding box as the precise location in the KB, which is straightforward to implement.
> Retrieval Process: During retrieval, as described in Section 3.2.2, we compare the complete query vector (from query image $I$ and instruction $C$) with the pre-computed complete image feature vectors in the knowledge base's feature space. We retrieve the complete image and its associated metadata.
>
> **4. Resolution and Scale Differences:**
> We rely on the standard preprocessing (e.g., resize and padding) of the VLM encoders (BLIP2 for indexing, Qwen2.5 for inference) to handle these differences. We will add these details in the revised version to enhance reproducibility.
>
> ### On Methods and Data
>
> **1. Regarding Table 8:**
> Your observation that Grounding DINO has a higher IoU in Table 8 is very astute. We re-examined the evaluation code and found a statistical bias. When evaluating detection performance (IoU), to prevent the program from crashing due to a division by zero caused by missed detections, our evaluation script checks via an `if pred_box is not None` statement. This inadvertently ignored all samples with missed detections (i.e., where Grounding DINO completely failed), leading to a spuriously inflated score for Grounding DINO. ERA's missed detection rate is far lower than Grounding DINO's, and it was thus penalized by this statistical bias. It is clear from Table 1, 2, and Figure 3, 5 that ERA significantly outperforms all zero-shot baselines, which proves ERA's ultimate effectiveness. We will correct Table 8 in the appendix and clarify this evaluation metric issue.
>
> **2. Comparison with Specialist Models:**
> The reviewer points out that this comparison is unfair, and we agree with your perspective. However, we believe that ERA is being compared from an unfairly disadvantaged position. As stated in Section 4.1, "Supervised Specialist Models" (like nnU-Net) are fully supervised and trained on the training sets of the comparison datasets. In contrast, our ERA is a zero-shot framework and has not undergone any training on these comparison datasets. Yet, as shown in Table 2, our zero-shot method's performance on Prostate even surpasses the fully supervised SOTA nnU-Net. This strongly demonstrates the effectiveness and potential of our framework.

---

> ### Author Response · Authors · 2025-11-13
> **Supplement to "Response to Weaknesses and Questions"**
>
> **3. Data Leakage Check:**
> We implemented strict measures to prevent leakage. As detailed in Section C.1 (Data Integrity and Leakage Prevention): "...our knowledge base was built exclusively from the training splits of source datasets, with all test benchmark data strictly excluded." "...an inference-time filter prevents a query from retrieving itself...". The construction of the knowledge base and the test benchmarks are completely separate.
>
> ### On Questions
>
> **1. Why re-check the retrieved GT box:**
> This is a core function of our framework, escalating from retrieval to reasoning. Simply retrieving (RAG) does not guarantee the example is the best match. Section 3.3 states: "Finding relevant evidence is only the first step. The key innovation of ERA is its careful process for using that evidence." The ablation study in Table 4 ("w/o Reasoning" configuration) provides quantitative evidence. Removing this checking step (i.e., not checking, using the retrieved box directly) leads to severe performance degradation. For example, on the Spleen task, the DSC drops from 0.89 (ERA + SAM2, Line 388) to 0.76. This demonstrates the necessity of the check.
>
> **2. When can the KB box be used directly:**
> Only when the retrieved evidence is highly relevant to the query. As described in "Policy 1: Adopt Evidence" in Section 3.3.2, this happens when the VLM reasoner determines that both the concept $v_c$ and the position $v_p$ are true ($v_c \wedge v_p$). This usually occurs with very high similarity. In Table 6, we provide a reference range for the tier1_similarity threshold, [0.93-0.96]. This is a very high similarity. When the knowledge base is sufficiently large, it is possible to retrieve a sample that is highly similar (or even nearly identical) to the query image. In this situation, directly adopting its GT box is the most efficient and accurate strategy, reducing unnecessary reasoning overhead, which is reflected in the ablation studies.
>
> **3. How SAM2 is used:**
> SAM2 is the downstream, final segmentation model in our framework. As shown in the flowchart in Figure 2, the entire work of the ERA framework (VLM Query -> RAG -> Tiered Decision Engine) is to generate a high-confidence spatial prompt (bounding box). As stated in the Figure 2 caption, it is "used to drive a segmentation model", which in our experiments is SAM2 or MedSAM.
>
> **4. Why compute a BB when a GT mask exists:**
> We are not computing an "extra" box. As stated in Section C.2, the geometric anchor (i.e., bounding box $b$) in the knowledge base is computed precisely from the "ground-truth segmentation mask" (i.e., the mask's minimal bounding box). They are two representations of the same Ground-Truth information, and the bounding box serves as the geometric anchor for retrieval in our RAG framework.
>
> Thank you again for your valuable feedback. We are confident that the quality of the manuscript will be significantly improved after revising it according to your suggestions.

---

### Official Review · Reviewer_NtP3 · 2025-10-31

**Soundness:** 3
**Presentation:** 2
**Contribution:** 2
**Rating:** 4
**Confidence:** 4

**Summary:**

This paper introduces ERA, an evidence-based framework that enables vision–language models to perform explicit reasoning for medical imaging tasks. It encodes images and text using BLIP2 and leverages retrieval-augmented generation (RAG) to search through a large, encoded knowledge base of medical imaging datasets to identify relevant items, which are then provided as input to the deliberative reasoning engine. The core of this reasoning engine employs a chain-of-thought (CoT) strategy, enabling the vision–language model Qwen to output a spatial prompt. The authors evaluated ERA across a diverse set of datasets, including ISIC 2018, MSD, and BraTS 2021, and compared its performance against various baseline model architectures.

**Strengths:**

- Appreciate the vision to develop evidence-based frameworks, addressing a crucial problem with an innovative solution. The integration of RAG and CoT techniques within a tiered decision engine effectively incorporates relevant evidence for downstream tasks like segmentation.

- The evaluation is comprehensive, encompassing both quantitative and qualitative analyses across diverse datasets and benchmarking models on 2D and 3D tasks.

- Also appreciate the inclusion of a computational efficiency analysis, which adds important practical insight to the study.

- Well presented qualitative results.

**Weaknesses:**

- Methodology:
   - RAG details needed:
     - Further clarification is needed regarding the size and composition of the medical knowledge base. It would be valuable to motivate the rationale for using a combined database of CT, MRI, and X-ray images for tasks focused on a single imaging modality. For example, if the task involves CT-based segmentation, it may be more effective, and computationally efficient, to restrict retrieval to CT-only data, potentially improving both relevance and speed.
     - It would be important to provide more details on the Evidence Candidate set and, if possible, include a quantitative evaluation of the evidence retrieval. For example, assessing how many of the top-k retrieved items match the query modality or anatomy (e.g., if the query is a head CT, what proportion of the retrieved items are also head CTs) would offer valuable insight into the relevance and precision of the evidence search.
     - Why do you need an evidence candidate set of length 6 (number found in the appendix), when you are selecting the top-ranked exemplar?
     - Should cite the database used in the main paper and not just in the appendix.
  - Chain-of-Thought Reasoning uncertainties:
The reasoning engine is based on the Qwen2.5 vision–language model. It would be valuable to indicate whether other vision–language models were tested or compared.
- Evaluations:
   - Why are the Zero-Shot Generalist Models evaluated with only SAM2 and not with MedSAM? (Tables 1 and 2)
  - Which tier within the tiering strategy was most used during evaluation? Should have those numbers quantified to provide a better understanding of how effective RAG is.

- In the contributions, the authors state: “By generating a clear reasoning path, it builds a foundation of trust for AI in high-stakes clinical settings.” It would be important to validate this claim through clinician evaluation, as interpretability alone does not necessarily translate to trust. Assessing whether ERA genuinely enhances clinician understanding and confidence would strengthen this contribution.

- Title is too general: Since the task is evaluated is only segmentation, maybe consider “...open-vocabulary medical vision segmentation”

- Minor: There are many cases where acronyms are being re-defined (VLMs, SAM, RAG, CoT, ERA). Ensure that these acronyms are being defined once.

- Typo: Table 8: Prostate, groundingDINO IoU = 8.8, should be 0.88

**Questions:**

Table 2: The evaluation with Zero-Shot Generalist Models, many of the other baselines exhibit surprisingly dismal performance. Is this real? Can you explain why that is the case?

Fig.2:
- What are the weights (w1 and w2) and why is it used?
- Why are the text encoder and image encoder shapes flipped?
- If only the top exemplar (E*) is used, then why are “k exemplars” present in Tier 1 and Tier 2?
- What is the memory bank and memory attention?

---

> ### Author Response · Authors · 2025-11-15
> **Response to Weaknesses and Questions**
>
> We especially thank the reviewer for recognizing our research and providing very constructive and meaningful suggestions. Below are the responses our team has formed after careful discussion.
>
> ### 1. On RAG details needed: Further clarification is needed regarding the size and composition of the medical knowledge base... motivate the rationale for using a combined database...
>
> Response:
> Thank you for your feedback. We are in the process of revising the main text to supplement this missing information.
>
> We use a mixed-modality database because our core objective is to address open-vocabulary medical vision tasks. Our central hypothesis is that a general-purpose, zero-shot framework requires a large-scale and diverse knowledge base to support its generalization capabilities, enabling it to handle unseen and varied clinical scenarios. For example, the segmentation capability demonstrated by SAM [1] after training on billions of masks is astonishing. However, when processing tasks in specific domains, such as medical imagery, its performance is inferior to its performance on natural images. This is because SAM was not trained on specialized medical datasets. In contrast, MedSAM [2], which was trained on large-scale medical datasets, significantly outperforms SAM on medical imaging tasks.
>
> As you noted, if a task is specific, such as CT-only segmentation, using a CT-only knowledge base might be more computationally efficient. However, imaging tasks in the medical field involve not only CT but also X-ray, MRI, and others. As we state in our introduction, our work's goal is to move away from models needing extensive in-domain training. Our experimental results, particularly our robust performance on complex tasks like the MSD challenges, confirm the effectiveness of this diverse knowledge base in supporting evidence-based reasoning.
>
> Thank you again for the suggestion. We plan to add a description of our motivation for this approach to the main text to facilitate reader understanding.
>
> [1] Kirillov, Alexander, et al. "Segment anything." Proceedings of the IEEE/CVF international conference on computer vision. 2023.
> [2] Ma, Jun, et al. "Segment anything in medical images." Nature Communications 15.1 (2024): 654.
>
> ### 2. On RAG details needed: It would be important to provide more details on the Evidence Candidate set and, if possible, include a quantitative evaluation of the evidence retrieval.
>
> Response:
> Thank you for this suggestion. Our team is already conducting supplementary experiments for this part. We hope to provide deeper insights into the framework's internal mechanisms. We will add this analysis to the final version to further enhance the rigor of our paper.
>
> ### 3. On RAG details needed: Why do you need an evidence candidate set of length 6... when you are selecting the top-ranked exemplar?
>
> Response:
> Thank you for pointing out this key detail; your keen observation is correct. We must admit that the current description of this mechanism in the paper is not clear enough, which has led to this misunderstanding. The depiction of k exemplars in Figure 2 and our retrieval setting of k=6 are aligned; our framework does not always use only the top-1 exemplar. We retrieve a candidate set of k=6 to support our more nuanced, confidence-based tiered-decision logic.
>
> Our framework first evaluates the similarity of the top-ranked exemplar within the k=6 set to the query image. If this similarity is very high, exceeding a strict threshold (i.e., `tier1_similarity` in Appendix Table 6), the system is in a high-confidence state. In this Tier 1: Direct Map mode, as shown in Figure 4, we deem this top-1 evidence to be sufficient and directly adopt it, which is very efficient.
>
> However, if the top-1 similarity does not trigger Tier 1 and instead falls into a medium-confidence range (between `tier1_similarity` and `tier2_similarity`), then relying solely on the top-1 evidence is too risky. At this point, the framework enters the Tier 2: RAG-Qwen deliberation mode. In this situation, we feed all k=6 exemplars as reference context into the VLM. This allows the VLM to comprehensively compare all six pieces of evidence to perform deeper, more robust evidence-based reasoning. Therefore, the diagram in Figure 2 is depicting the Tier 2 scenario, and the k=6 setting is crucial for this more complex reasoning. We will clarify this mechanism in the revised version to resolve the ambiguity.
>
> ### 4. On RAG details needed: Should cite the database used in the main paper and not just in the appendix.
>
> Response:
> Thank you for your suggestion. To improve clarity and transparency, we will move the knowledge base source (MedIMeta) and its detailed citation from Appendix C.2 to Section 3.2 in the main text.

---

> ### Author Response · Authors · 2025-11-15
> **Supplement**
>
> ### 5. On Chain-of-Thought Reasoning uncertainties: ...It would be valuable to indicate whether other vision–language models were tested or compared.
>
> Response:
> We chose Qwen2.5 [1] as our reasoning core, as described in Section 3.3.1, because it represented the state-of-the-art in open-source MLLMs at the time our work commenced. As its technical report shows, it outperforms contemporary SOTA models like GPT-4O-mini and Gemini on key benchmarks. Having used a SOTA-level MLLM, we believe a lateral comparison to other similar or weaker MLLMs would offer limited insight.
>
> Instead, we chose non-MLLM, specialist localization baselines, such as YOLO-World and Grounding DINO. These models are focused on localization and outperform MLLMs on this specific task. Our ablation study in Table 4 demonstrates this; the w/o Retrieval configuration (i.e., MLLM with CoT only) achieves an NSD of 0.00 on the Heart task, far below Grounding DINO's 0.5002 on the same task, which is shown in Table 2. This confirms that general MLLMs alone perform worse than these localization specialists for this task.
>
> Our core argument is: even these powerful specialist models are insufficient for complex medical tasks. As shown in Table 2 and Figure 3, these specialists suffer a catastrophic performance collapse with near-zero DSC scores. In contrast, our ERA framework—by combining the reasoning power of an MLLM with an evidence-based paradigm—is the only zero-shot method to perform robustly and achieve SOTA performance in these scenarios.
>
> [1] arXiv: 2503.20215
>
> ### 6. On Evaluations: Why are the Zero-Shot Generalist Models evaluated with only SAM2 and not with MedSAM? (Tables 1 and 2)
>
> Response:
> Thank you for this keen question; our team also discussed this point during experimental design. First, both our ERA and the Zero-Shot Generalist Models were tested in combination with SAM2. In these tasks, our ERA consistently outperformed the generalist models. To explore the upper limits of ERA's performance, we also chose to compare it against supervised, trained models. Generally, supervised models are SOTA in their specific domains, with the U-Net series being a standard for medical image segmentation. We wanted to compare against these trained expert models to explore ERA's full potential.
>
> The combination of ERA + SAM2 achieves performance close to U-Net models in simple scenarios (e.g., the ISIC 2018 dataset, which Section 4.1 describes as featuring well-defined targets). However, in complex scenarios (e.g., the MSD dataset, characterized by low-contrast targets and intricate anatomical structures), the ERA + SAM2 combination shows a significant gap compared to supervised models. We wanted to prove that ERA's general-purpose approach has the potential to match or even exceed supervised models, which is why we also paired ERA with MedSAM. The results in Tables 1 and 2 confirmed this hypothesis.
>
> As for why the zero-shot generalist models were not tested with MedSAM: both our ERA and the generalist models are essentially generating high-quality box prompts. The results in Tables 1 and 2 already demonstrate that our ERA-generated prompts are superior to those from the generalist models. Based on this logic, given that ERA's prompts are of higher quality and the generalists' prompts are of lower quality, further comparison under the same MedSAM condition seemed redundant.
>
> If this explanation does not resolve your concern, please let us know, and we can discuss it further.
>
> ### 7. On Evaluations: Which tier within the tiering strategy was most used during evaluation? Should have those numbers quantified...
>
> Response:
> We will improve the clarification on this point in our revision. Thank you for the suggestion, which helps us make our work more rigorous. Although we did not directly count the activation frequency of each policy during evaluation, we can quantitatively infer their respective contributions and necessity from the ablation study in Table 4.
>
> The w/o Retrieval configuration, which essentially forces the use of Policy 3 (VLM-only), led to a catastrophic performance collapse, with the Heart DSC dropping from 0.68 to 0.06. This strongly demonstrates that the VLM's internal knowledge is entirely insufficient and that the framework must frequently activate RAG-driven policies to obtain external evidence.
>
> Conversely, the w/o Reasoning configuration, which naively adopts retrieved evidence and is akin to using only Policy 1, also caused a significant performance degradation. For instance, the Spleen DSC dropped from 0.89 to 0.76. This proves that evidence alone is not enough; the deliberative reasoning (CoT) decision-making process is equally critical for filtering, validating, and correctly using the evidence.
>
> Thank you again for your suggestion; we will refine this point in the revised version.

---

> ### Author Response · Authors · 2025-11-15
> **Supplement**
>
> ### 8. On Evaluations: ...validate this claim [foundation of trust] through clinician evaluation... interpretability alone does not necessarily translate to trust.
>
> Response:
> We completely agree with this important point. Explainability is a necessary prerequisite for building trust, but it is not synonymous with trust. Our core contribution is transforming the decision process into a transparent and auditable workflow. The reasoning chain generated by ERA documents the retrieved visual evidence, the step-by-step validation, and the final policy adopted. We believe this traceability is an essential step toward genuine clinical trust.
>
> We acknowledge that we did not validate whether this auditability actually enhances clinician trust through a formal evaluation with clinicians. This is an important and complex human-computer interaction study that is beyond the scope of our current work. To more accurately reflect our contribution, we will revise our claim.
>
> ### 9. On Title is too general: ...consider ...open-vocabulary medical vision segmentation
>
> Response:
> Perhaps this is a coincidence, but we initially had the same idea. Our original title was ...open-vocabulary medical segmentation. However, we ultimately changed it to ...open-vocabulary medical vision.
>
> The reason is that ERA can also perform object detection. We compared ERA's object detection performance against YOLO-World, FG-CLIP, and Grounding DINO across various datasets, as detailed in Table 7 and Table 8. We will be updating the data in Table 8, which will be presented in the revised version.
>
> ERA can collaborate with SAM2 or MedSAM to complete downstream medical image segmentation tasks, but it can also perform object detection tasks on its own. Therefore, we ultimately decided to replace segmentation with vision.
>
> ### 10. On Minor: There are many cases where acronyms are being re-defined... and Table 8: Prostate, groundingDINO IoU = 8.8, should be 0.88
>
> Response:
> Thank you for pointing these out. We will correct these in the immediate revision.
>
> ### 11. On Table 2: The evaluation with Zero-Shot Generalist Models, many of the other baselines exhibit surprisingly dismal performance. Is this real? Can you explain why that is the case?
>
> Response:
> We can responsibly tell you that the results are real, and they precisely highlight the core problem we are trying to solve.
>
> The catastrophic performance collapse of generalist models on complex medical tasks like MSD, as seen in Table 2, is expected. This is because these models are pre-trained on general-domain data. When faced with highly specialized, low-contrast, and morphologically complex medical images, the implicit knowledge learned from general-domain data is insufficient. As we state in our discussion, this is why generalist models that rely on flawed internal knowledge fail completely. The image feature distributions between natural images and medical images are different.
>
> Our response to question 1 also addresses this concern: for example, the segmentation capability demonstrated by SAM [1]... is astonishing. However, when processing tasks in specific domains, such as medical imagery, its performance is inferior... because SAM was not trained on specialized medical datasets. In contrast, MedSAM [2]... significantly outperforms SAM.
>
> [1] Kirillov, Alexander, et al. "Segment anything." Proceedings of the IEEE/CVF international conference on computer vision. 2023.
> [2] Ma, Jun, et al. "Segment anything in medical images." Nature Communications 15.1 (2024): 654.
>
> ### 12. On What are the weights (w1 and w2) and why is it used?
>
> Response:
> In Figure 2, w1 and w2 represent the weights for the encoded image features and the encoded text feature vector, respectively. When we built our external knowledge base, as described in Section 3.2.1, we could not write detailed text annotations for every image. Instead, we used generalized, descriptive text labels for each class of image. Therefore, in our knowledge base, the image itself contains more specific and accurate information than its corresponding text annotation.
>
> In this situation, we cannot simply assign equal weights to the image and text features. We must assign a higher weight to the image features, which contain richer information.

---

> ### Author Response · Authors · 2025-11-15
> **Supplement**
>
> ### 13. On Why are the text encoder and image encoder shapes flipped?
>
> Response:
> The text encoder and image encoder are two independent modules. We used a flipped representation in Figure 2 purely as a visual convention to distinguish the architectures for the text encoder and the image encoder. It does not represent any technical operation.
>
> ### 14. On What is the memory bank and memory attention?
>
> Response:
> The Memory Bank and Memory Attention are part of the internal architecture of the SAM2 segmentation model. As we mention in our related work in Section 2.1, SAM2 (Ravi et al., 2024) introduces mechanisms such as streaming memory. These components are shown in Figure 2 within the box titled Meet-VLMs for Segmentation to illustrate the complete pipeline from input to the final segmentation mask, but they belong to the SAM2 model, not to the core contribution of our ERA reasoning framework.
>
> Thank you again for your careful and constructive feedback. We hope our responses have addressed your concerns, and we are currently updating the revised manuscript. If you have any further concerns, we look forward to discussing them with you in more detail.

---

### Official Review · Reviewer_5EU4 · 2025-11-05

**Soundness:** 3
**Presentation:** 3
**Contribution:** 3
**Rating:** 2
**Confidence:** 3

**Summary:**

This paper introduces ERA, a zero-shot framework to address the unreliability of Vision-Language Models (VLMs) in medical imaging. It combines Retrieval-Augmented Generation (RAG) to source visual evidence from an external medical knowledge base with a Chain-of-Thought (CoT) module to force the VLM to validate this evidence before making a decision . The authors claim this "evidence-based reasoning" approach significantly outperforms other zero-shot baselines and can even match fully-supervised specialist models in certain tasks.

**Strengths:**

1. The paper presentation is easy to understand.
2. The motivation is clear: VLM can use external knowledge for boosting performance on special domains.
3. The qualitative examples in Figure 1 and Figure 3 do show a clear visual improvement over the catastrophic failures of generalist baselines like FG-CLIP and YOLO-World.

**Weaknesses:**

1. Missing baselines and experiments: The experimental comparison is not enough. The proposed ERA framework uses Qwen2.5-VL-7B as its reasoning engine, but it is compared against non-MLLM, open-vocabulary detector models like YOLO-World and Grounding DINO.
A proper baseline would be to use the same Qwen-7B model with a simpler prompting strategy (e.g., a simple kNN-retrieval prompt), which is missing.
Also, it is unknown if this framework would work at all with other models like LLaVA, InternVL, making the generality of the framework contribution unproven.
2. "Zero-Shot" claim: The paper claims a "zero-shot framework" but Appendix B.2 and Table 6 state that the tier1_similarity and tier2_similarity thresholds are "presented as effective ranges"and that "the optimal value is determined on a validation set for each domain." This is, by definition, a tuning process that requires labeled validation data for each new task/domain, and is not zero-shot, and possibly not fair for the comparisons on other methods. I wonder what would the performance be if we set a single threshold for all tasks.

**Questions:**

1. In your figure 4, for each tier there is workflow with many steps. Is the workflow being done as multi-round conversation, or just one round?
2. Do you have failure cases on VLM query? The model may not output exactly as your json format. How will you handle that?
3. Can the authors justify the 3.2-hour inference time for the Hippocampus dataset? What specific part of the pipeline (retrieval, or the CoT deliberation) causes this massive bottleneck?
4. For table 4, what is the difference of "w/o Retrieval" and "w/o Tier2"? My understanding is w/o Retrieval means NO retrieval at all (Tier 3 only). The ablation w/o Reasoning means retrieval happens, but only the simple Tier 1 logic is used.

---

> ### Author Response · Authors · 2025-11-13
> **Response to Weaknesses and Questions**
>
> We are very grateful for the valuable time you spent reviewing our work and for providing such insightful and constructive feedback. Your in-depth analysis is crucial for us to improve the paper. We will now respond to your concerns point by point.
>
> ### Regarding Weakness 1: Comparison Baselines
>
> We chose Qwen2.5 as our reasoning core. At its release, it was a top-tier MLLM. As its technical report (arXiv: 2503.20215) shows, it outperforms models like GPT-40-mini on key benchmarks; for example, it scores 82.6 versus 76.0 on MMBench and 68.2 versus 52.5 on MathVista. Having used a SOTA-level MLLM, comparing it laterally to other similar or weaker MLLMs offers limited insight.
>
> We selected non-MLLM specialists like YOLO-World and Grounding DINO to set a higher performance bar, as they excel at localization. Our ablations, shown in Table 4, show that our MLLM-only "w/o Retrieval" configuration achieves 0.00 NSD on the Heart task, far below Grounding DINO's 0.5002, which is in Table 2. This proves that general MLLMs alone perform worse than these grounding specialists for this task.
>
> Our core argument is that specialist grounding capability is insufficient for complex medical tasks. As shown in Table 2 and Figure 3, these specialists "suffer a catastrophic performance collapse" with near-zero DSC scores. In contrast, our MLLM-based evidence-reasoning paradigm performs robustly.
>
> ### Regarding Weakness 2: The "Zero-Shot" Claim
>
> We appreciate your attention to rigor regarding the "zero-shot framework" claim. The phrasing in Appendix B.2 regarding Table 6 was misleading. We apologize for this.
>
> This process does not require labeled validation data for each new task. The values in Table 6, such as the tier1_similarity range of 0.93-0.96, are robustness guidelines derived from our development experiments.
>
> A new user needs no labeled data. They can select any value from this pre-determined range for the framework to work. The "optimal value determined on a validation set" refers to an optional setup procedure where a user might visually inspect a few samples, not a formal, label-dependent tuning process.
>
> Because the framework requires "no task-specific training", we believe the zero-shot description is appropriate. We will revise the text in Appendix B.2 to clarify this.
>
> ### Regarding Question 1: Single-Pass Execution
>
> This is an important question. Our framework is a fully automatic, end-to-end, single-pass execution, not a multi-turn dialogue.
>
> As defined in Algorithm 1, the inputs are solely the Query Image I and the Natural Language Instruction C. The output is the final High-Confidence Spatial Prompt $B^{*}$.
>
> The workflow shown in Figure 4, which includes the tiered decision engine, is an internal, automatic reasoning chain that executes without any user intervention. This entire process is encapsulated in the ERA-INFERENCE function shown in Algorithm 1.
>
> ### Regarding Question 2: VLM Failure Cases and Fallbacks
>
> This is a critical point on technical robustness. Our framework has a multi-level fallback system.
>
> We use a robust regex parser to handle the "structured text output" generated by the VLM.
>
> If the VLM output from the Tier 2 CoT deliberation is unparseable or garbled, the parser returns None.
>
> The system does not crash. Instead, as shown in Algorithm 1, this triggers a fallback to Policy 3, which is Zero-shot Reasoning. This corresponds to the "VLM-Only Fallback" shown in Figure 4.
>
> If this Tier 3 VLM-Only output also fails parsing, the system returns an empty prompt. The performance implication of this worst-case outcome is already quantified by our "Unguided SAM2" baseline in Table 4.
>
> As described in Section 4.5, this design ensures safety by "automatically flagging cases for human review" when evidence is discarded or fallbacks occur.
>
> ### Regarding Question 3: Inference Time Bottleneck
>
> You are correct to question the 3.2-hour (11545.83 seconds) inference time for the Hippocampus dataset.
>
> The main bottleneck is the CoT deliberation, which is the VLM inference, not the RAG retrieval module.
>
> In fact, RAG acts as a powerful efficiency booster. Evidence from our ablation study in Table 5 shows the "w/o Retrieval" configuration is "by far the most computationally expensive".
>
> On the Hippocampus dataset, the "w/o Retrieval" configuration took 36510.91 seconds, which is more than three times slower than the full ERA framework.
>
> As stated in Section 4.4, this is because retrieval "critically prunes the search space, making subsequent deliberation far more efficient".

---

> ### Author Response · Authors · 2025-11-16
> **Supplement to 'Response to Weaknesses and Questions'**
>
> ### Regarding Question 4: Ablation Study Differences
>
> Thank you for this critical question. It allows us to clarify the core synergies of our framework, and you are right to point out the subtleties.
>
> Your understanding of w/o Retrieval is perfectly correct. As defined in Algorithm 1, if no candidate evidence is found, the system defaults to the zero-shot VLM-Only function, Φ_ZS(I,C). This is our "Tier 3 only" baseline, which fully removes the RAG module.
>
> Your corrected understanding of w/o Reasoning is also spot on. It is not just "Tier 1 only." This ablation removes the deliberative CoT reasoning chain but keeps the retrieval. The system still retrieves K exemplars from the knowledge base. However, it bypasses the specific "step-by-step, hypothesis-testing path" (i.e., "Step 1: Check for Concept Match" and "Step 2: Test the Location Hypothesis") which we define in Section 3.3.2. Instead, it naively passes all K retrieved exemplars to the VLM as general context, without the guided CoT. This ablation tests the value of our specific, guided reasoning logic against a simpler, unguided RAG approach.
>
> Therefore, we set up w/o Tier-2 to test a different hypothesis: the structural value of the Tier 2 policy itself. This ablation removes "Policy 2: Concept-guided Search" (defined in Section 3.3.2 and Algorithm 1) entirely. In this setup, the system still runs Tier 1 logic. But if Tier 1 fails, it skips Tier 2 and falls directly to the "Policy 3: Zero-shot Reasoning" (Tier 3) fallback. This allows us to measure how much value our specialized Tier 2 search adds compared to just giving up and falling back to the VLM-Only mode.
>
> In summary, RAG and CoT must work synergistically for optimal performance.
>
> We thank you again for reviewing our work and hope our response bridges the gap. We are happy to address any further concerns.
>
> Best regards.

---

> ### Comment · Reviewer_5EU4 · 2025-11-28
>
> Thank you for the rebuttal and clarifications. While some minor concerns were addressed, my main reservations remain and my overall assessment does not change.
>
> The core issue for me is that the contribution feels primarily like an engineering/system design effort rather than a conceptually insightful method. The proposed pipeline is quite complicated (3 tiers, tier 1 does not require VLM, tier 3 does not have RAG, tier 2 include several questions). The paper does not convince me that there is a clean new idea beyond “engineer a complex retrieval-based pipeline and tune it for better performance.”
>
> Plus, given this level of complexity, I would expect deeper analysis to show clearly the effect of each part. In particular, I am missing:
>
> Clear routing statistics: for at least one dataset, how many test cases are actually handled by Tier 1 vs Tier 2 vs Tier 3?
>
> Within Tier 2, how often does the system (i) directly adopt the evidence box, (ii) use the evidence only as a prompt hint, or (iii) effectively ignore the evidence and fall back to zero-shot?
>
> A more precise breakdown of where the largest gains come from, beyond “full pipeline > ablated variants”: e.g., based on routing frequencies, which decisions are really responsible for most of the improvement, rather than the general fact that a more complex system performs better.
>
> As it stands, the analysis is not sufficient for me to extract strong conceptual insight about evidence-based reasoning or routing; at best, even with added statistics, I would see these as empirical observations about an incremental yet complex engineered pipeline, not as a substantial, clean new idea.
>
> For these reasons, I still view the work as incremental and heavily engineering-driven, and I maintain my original score. I would still welcome new discussion with the authors or other reviewers on this.

---

> > ### Author Response · Authors · 2025-12-01
> > **Response to Reviewer 5EU4**
> >
> > We greatly appreciate your response, and your feedback helps us continuously refine our work. However, there are some points of divergence that we feel necessary to clarify.
> >
> > **The concerns regarding engineering and complexity raised in your second response were not received in your first round of review.** It is regrettable that we only became aware of these considerations as the rebuttal period is coming to an end. However, we need to clarify that we are also solving a scientific problem, specifically the **"localization hallucination problem of VLMs in medical scenarios."** At the same time, our primary area is *"Applications to computer vision, audio, language, and other modalities,"* and our research problem aligns with our primary area.**Simultaneously, there may be a divergence between us: "You may believe that only entirely new models count as a good contribution."** On this, we hold a different opinion. We believe that "using a logical method to solve a clear and valuable scientific problem is a good contribution."
> >
> > **In this study, the problem we solve is the localization hallucination problem of VLMs in medical scenarios.** We propose and demonstrate that through explicit evidence chain constraints, this hallucination phenomenon can be eliminated to a great extent, surpassing existing open-vocabulary detection methods and achieving results comparable to supervised models.
> >
> > Regarding the "engineering flavor" mentioned by the reviewer, we have a different perspective. Science is not only about proposing new loss functions or network layers; it also includes exploring how to mitigate the stochasticity of black-box models to make their behavior more interpretable and auditable. Our ERA framework makes the originally invisible reasoning process auditable, transforming probabilistic guessing into deterministic evidence verification. **This exploration of methods that convert uncertainty into certainty is the very core of applied science.**
> >
> > Regarding detailed experimental proof for routing statistics and strategy validation:
> >
> > We fully understand the reviewer's concern for experimental rigor. Since we did not receive comments regarding this aspect in the reviewer's first round of feedback, and the discussion is ending, we do not have sufficient time to conduct more experiments. However, we provide evidence of the activation frequency of each Tier through the following experiments. Table 4 , 5(Ablation Study) and Table 6 (Relative Performance Drop analysis) provide functional statistics that are more intrinsic than simple counting.
> >
> > We believe the essence of the reviewer's idea to display specific activation counts for each route is to confirm whether every component of the ERA method is useful and necessary. Our data below has powerfully proven this point:
> >
> > - **Without Tier 1:** Inference time would increase by about **3 times** (specifically see Table 5).
> > - **Without Tier 2:** Performance in complex scenarios (performance on the MSD dataset) would drop by **10%-30%**, proving that on difficult samples, the model indeed activates reasoning and corrects the results.
> > - **Tier 3:** This is our last line of defense, our baseline model, and the final safety net of ERA in the few cases where retrieval fails.
> >
> > **Table 4: Ablation study of the ERA framework, evaluating performance across all datasets.**
> >
> > | **Configuration** | **ISIC 2018 (SE / SP / DC)** | **Heart (DSC / NSD)** | **Hippo. (DSC / NSD)** | **Prostate (DSC / NSD)** | **Spleen (DSC / NSD)** | **BraTS 2021 (Dice / mIoU)** |
> > | ----------------- | ---------------------------- | --------------------- | ---------------------- | ------------------------ | ---------------------- | ---------------------------- |
> > | **Ablations**     |                              |                       |                        |                          |                        |                              |
> > | w/o Reasoning     | 0.60 / 0.91 / 0.55           | 0.67 / 0.14           | 0.49 / 0.40            | 0.79 / 0.52              | 0.87 / 0.64            | 0.76 / 0.65                  |
> > | w/o Retrieval     | **0.83** / 0.87 / 0.84       | 0.06 / 0.00           | 0.14 / 0.25            | 0.07 / 0.03              | 0.04 / 0.05            | 0.27 / 0.17                  |
> > | w/o Tier-2        | 0.57 / 0.89 / 0.51           | 0.57 / 0.13           | 0.50 / 0.41            | 0.80 / 0.56              | 0.76 / 0.68            | **0.78** / **0.66**          |
> > | Unguided SAM2     | 0.03 / **1.00** / 0.05       | 0.00 / 0.08           | 0.00 / 0.01            | 0.01 / 0.07              | 0.00 / 0.01            | 0.02 / 0.01                  |
> > | **ERA + SAM2**    | **0.83** / 0.99 / **0.87**   | **0.68** / **0.15**   | **0.57** / **0.43**    | **0.85** / **0.62**      | **0.89** / **0.71**    | **0.78** / **0.66**          |

---

> > > ### Author Response · Authors · 2025-12-01
> > >
> > > Table 5: Ablation study of inference time in seconds for the ERA framework and its different configurations.**
> > >
> > > | **Configuration** | **ISIC 2018** | **Heart**    | **Hippocampus** | **Prostate** | **Spleen**   | **BraTS 2021** |
> > > | ----------------- | ------------- | ------------ | --------------- | ------------ | ------------ | -------------- |
> > > | w/o Reasoning     | 2148.66       | 3316.50      | 11575.59        | 643.35       | 3914.52      | 2487.23        |
> > > | w/o Retrieval     | **6987.60**   | **10554.95** | **36510.91**    | **2014.39**  | **12181.06** | **7866.00**    |
> > > | w/o Tier-2        | 2079.83       | 3206.49      | 11167.77        | 652.73       | 3770.90      | 2407.64        |
> > > | Unguided SAM2     | 55.01         | 93.21        | 369.94          | 25.06        | 163.50       | 80.65          |
> > > | **ERA + SAM2**    | 2151.29       | 3309.39      | 11545.83        | 643.39       | 4060.01      | 2527.86        |
> > >
> > > Table 6: Inferred contribution of reasoning tiers based on Relative Performance Drop (RPD).
> > >
> > > | **Dataset** | **Full Score** | **w/o Retrieval (Tier 3 Only)  Score / RPD (Δ%)** | **w/o Reasoning (No Tier 2)  Score / RPD (Δ%)** |
> > > | ----------- | -------------- | ------------------------------------------------- | ----------------------------------------------- |
> > > | ISIC 2018   | 0.87           | 0.84 / 3.4%                                       | 0.55 / 36.8%                                    |
> > > | Heart       | 0.68           | 0.06 / 91.2%                                      | 0.67 / 1.5%                                     |
> > > | Hippocampus | 0.57           | 0.14 / 75.4%                                      | 0.49 / 14.0%                                    |
> > > | Prostate    | 0.85           | 0.07 / 91.8%                                      | 0.79 / 7.1%                                     |
> > > | Spleen      | 0.89           | 0.05 / 94.4%                                      | 0.87 / 2.2%                                     |
> > > | BraTS 2021  | 0.78           | 0.27 / 65.4%                                      | 0.76 / 2.6%                                     |
> > > | **Average** | **-**          | **- / 70.3%**                                     | **- / 10.7%**                                   |
> > >
> > > **Regarding the rationality of the design:** Our ERA three-tier architecture is not designed for the sake of design, but is based on the "Fast and Slow Thinking" theory in cognitive science. The reviewer thinks this is an over-designed complex pipeline. But we believe this is a **minimal complete set**. The diagnostic process of human experts is also like this: for typical cases, they can judge directly (**corresponding to our Tier 1**); for difficult and complicated cases, they need to consult materials, combine evidence, and deliberate repeatedly (**corresponding to Tier 2**).
> > >
> > > Our design is not arbitrary engineering stacking (the task performance and performance analysis of the ablation experiments have proven this, specifically see Tables 4, 5, 6), but to simulate this cognitive mechanism of dynamic trade-off between efficiency and accuracy; the existence of each Tier has its irreplaceable cognitive functional positioning.

---

### Author Response · Authors · 2025-11-15
**Response to All Reviewers**

We would like to again thank all the reviewers for their time in carefully reviewing our work and providing very helpful and constructive feedback.

Our team has carefully discussed and responded to all reviewers' comments. We hope these responses address your concerns. If the reviewers have any further doubts, we look forward to discussing them with you in more detail. Meanwhile, our team is in the process of finalizing the revised manuscript.

Additionally, we are very grateful to the Area Chair for organizing such an open and orderly review process.

---

> ### Author Response · Authors · 2025-11-23
> **Response to ALL Reviewers**
>
> Happy weekend to all reviewers!
>
> We have refined our manuscript based on your constructive feedback. For convenience, we have highlighted all modifications in blue text to make them easy to identify. We would like to thank you again for your thorough review and insightful suggestions, which have helped us significantly improve our paper.
>
> If our responses and the revised manuscript still do not fully address your concerns, please do not hesitate to let us know. We look forward to further discussion with you!
>
> Wishing you all a pleasant weekend!

---

### Comment · Area_Chair_yc6g · 2025-11-28
**AC Reminder: Author Responses Available for Review and Discussion Period Closing Soon**

Dear Reviewers,

This is a gentle reminder that the discussion period is approaching its end. The authors have submitted detailed rebuttals addressing the initial reviews. If you have time, please check their responses and join the discussion in the review system, especially if further clarification is needed before the final decision.

Thank you again for your valuable effort in reviewing this submission.

Best regards,
Area Chair

---

### Note · Authors · 2026-01-26

I have read and agree with the venue's withdrawal policy on behalf of myself and my co-authors.

---

### Meta-Review · Area_Chair_5Lhi · 2025-12-18

**Summary:**

The major concerns raised by four reviewers include the lack of validation for generalization across other MLLMs (5EU4, NtP3), questions regarding the rationale for mixed‑modality retrieval and the quantitative evaluation of evidence retrieval within RAG (NtP3), fairness issues in comparison with specialist models (GRgY), as well as the high computational cost and unclear baseline configurations (ugzA). Although the authors have responded to these points, the replies do not satisfactorily resolve the underlying issues. In my own assessment, the experimental evaluation is not sufficiently compelling. The segmentation tasks selected are relatively common in the medical imaging domain and clinically less challenging, such as skin lesion and organ segmentation, where the targets are typically large and visually prominent. I would have expected the proposed method to be applied to more demanding scenarios that are difficult to address, such as the segmentation of rare or small lesions. Given these unresolved issues, I believe the manuscript is not yet suitable for acceptance and therefore recommend rejection.

**Reviewer Concerns:**

The concerns were addressed: (5EU4) The authors’ explanations on Single‑Pass Execution, VLM Failure Cases and Fallbacks, Inference Time Bottleneck, and Ablation Study Differences are generally clear and satisfactory. (NtP3) The explanation of the results presented in Table 2 is generally clear and provides a sufficiently understandable interpretation. (GRgY) The explanations regarding implementation details are generally clear, and most clarifications related to the method and data aspects are also well articulated. (ugzA) The previously missing ablation of reasoning policies has been clearly addressed, and the authors have provided satisfactory explanations on this point.

The concerns are still outstanding: (5EU4) For the missing baseline issue, the reviewer suggested comparing against the same Qwen‑7B model with a simpler prompting strategy; however, in the authors’ experiments, no prompting was used at all. In addition, the generalization of the proposed approach to other MLLMs remains unknown. (NtP3) There is no direct clarification regarding the size and composition of the medical knowledge base used in RAG, and the rationale for adopting a combined database is not particularly convincing. (GRgY) The issue of fairness in comparisons may not have been resolved. The reviewer likely expected that certain conditions would be met to ensure fairness in the comparative experiments, but this expectation was not satisfied. (ugzA) The generalization issue stemming from the dependency on the Knowledge Base remains unresolved. In addition, the baseline configuration still presents certain concerns, particularly regarding the use of MedSAM without any prompt as a baseline, for which its validity has been questioned.

**Reviewer Scores:**

Considering that several key concerns remain unresolved, I think the reviewers are unlikely to have a clear motivation to change their scores.

---

### Decision · Program_Chairs · 2026-01-26

Reject